# Caffarelli Regularity and Hierarchical Phase Boundaries in Diffusion Models

## Abstract

Recent studies have shown phase-transition-like behavior in diffusion models, where a small perturbation of the initial Gaussian noise sample can cause an abrupt change in the generated image. The underlying mechanism of these transitions, however, remains theoretically underexplored. In this work, we investigate this phenomenon through the lens of the pullback metric on the latent space induced by the perceptual similarity between generated images. We observe a hierarchical emergence of phase boundaries: coarse boundaries appear in the early denoising steps, while finer boundaries progressively emerge within these regions as the denoising process advances. Moreover, we observe that diffusion distillation shifts boundary formation towards earlier denoising steps and reduces final complexity by decreasing the number of sharp boundaries. To provide a theoretical foundation, we follow the JKO scheme and approximate the reverse diffusion dynamics by a discrete-time sequence of quadratic-cost optimal transport maps between successive noisy marginals. We show that mode splitting forces the diffusion generative map to develop large Lipschitz constant. Using Caffarelli's regularity theory, we argue that these high-Lipschitz regions form contiguous sets, driven by the disjoint support of the real data distribution and giving rise to phase boundaries. We further note that the proposed theoretical framework does not depend on models' design, but describes the general properties of unimodal-to-multimodal diffusion mappings. This leads to an important practical implication: non-Lipschitzness of generative mapping is necessary for good mode coverage.

## 1 Introduction

Generative diffusion and flow-matching models achieve state-of-the art performance in image and video generation Ho et al. (2020); Dhariwal & Nichol (2021); Rombach et al. (2022); Wan et al. (2025). They synthesize data by numerically solving the reverse probability flow ODE (PF-ODE) associated with a learned time-varying vector field Song et al. (2021); Lipman et al. (2023). The learned vector field defines a generative mapping from a gaussian prior to the data distribution and dictates the properties of the generative mapping (regularity, smoothness, Lipschitz constant) which control the stability and consistency of the PF-ODE's (numerical) solution. This, in turn, determines the ability of the generative model to mimic a real data distribution.

Our approach to study the regularity of a generative mapping is to analyze its behaviour under continuous interpolations between two initial Gaussian samples. A regular and smooth map transforms two "close" latent samples into two "close" images. The notion of closeness is introduced through a metric in both latent and image spaces. The simplest approach is to consider the Euclidean distance for both spaces, and a more sophisticated approach is to employ feature extractors aligned with human perception on the image domain, such as CLIP Radford et al. (2021), DINOv2 Oquab et al. (2023) or Inception-V3 Szegedy et al. (2016). If these metrics detect that close gaussian latents are transformed into very distant images, we consider it a sign of irregularity of the generative map.

**Differential-geometric view on generative map** Given a trained model and a perceptual metric, the pullback metric on a latent space is commonly said to define the latent geometry (VAEs (Shao et al., 2018; Arvanitidis et al., 2018), GANs (Wang & Ponce, 2021), diffusion models (Park et al., 2023)). It was demonstrated that linear interpolation in GAN and VAE latent spaces closely approximates geodesic interpolation, so that the latent space admits a flat metric. The curvature of the pullback

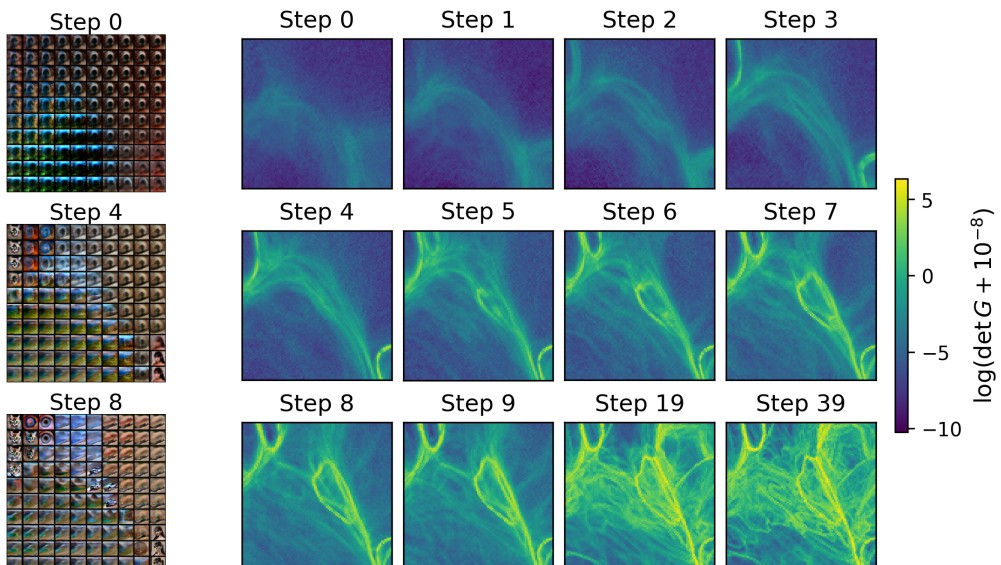

Figure 1: Hierarchical emergence of phase boundaries for Stable Diffusion 1.5: coarse boundaries appear in the early denoising steps, while finer boundaries progressively emerge within these regions as the denoising process advances. Left: a grid of predicted noiseless images, $\hat{x}_0(\alpha, \beta)$, over a 2D latent slice. Right: evolution of the determinant of the CLIP pullback metric, $G$, shown on a log scale; high values indicate regions where the features of the generated images change abruptly.

metric can be linked to the regularity of the generative map using classical result in differential geometry (Kobayashi & Nomizu, 1996): Any sufficiently smooth, globally invertible map $F : \mathbb{R}^d \to \mathbb{R}^d$ induces a pullback metric $G = (\nabla F)^T \nabla F$ that is flat. In short, by examining the pullback metric $G$, one gains an insight about irregularities of the generative map $F$.

**Disconnected support of data distributions** However, there is a fundamental topological obstruction that limits the ability of smooth generative maps to model complex, multimodal data distributions: connectedness is preserved under continuous mappings. Thus, a single deterministic generator cannot realize a target distribution of real data with truly disconnected support. The latter is an assumption known as the "Union of Manifolds Hypothesis", first proposed by Brown et al. (2023), which was experimentally verified on a wide range of real image datasets. Consequently, a smooth generator must either leak probability mass through low-density "gaps" or suffer from *mode-collapse*, a well-documented phenomena in GAN training (Kossale et al., 2022).

**Phase transitions in diffusion models** Diffusion models are known for superior mode coverage relative to GANs (Dhariwal & Nichol, 2021). However, as generation converges toward the data distribution, studies have documented phase transitions: temporal ones along the trajectory (Biroli et al., 2024; Yu & Huang, 2025) and spatial ones across the latent/data domain (Humayun et al., 2025; Lobashev et al., 2025; Yang et al., 2023).

**Our contributions** We summarize our key contributions as follows:

- We use the Jordan-Kinderleherer-Otto (JKO) scheme to approximate the reverse ODE by a composition of optimal transport (OT) maps, and apply Caffarelli regularity theory to each map via "cut trick", to explain emergence of phase boundaries in diffusion models.
- We prove that high Lipschitz constant regions of the diffusion generative map are associated with mode-splitting, thereby giving rise to phase boundaries.
- We empirically study the formation of boundaries in Stable Diffusion 1.5 and Wan 2.1 via pullback metric induced by perceptual similarity of images and observe hierarchical appearance of phase boundaries.
- We examine the effect of diffusion distillation on the formation of boundaries, and empirically show that the boundaries shift to the earlier diffusion steps.

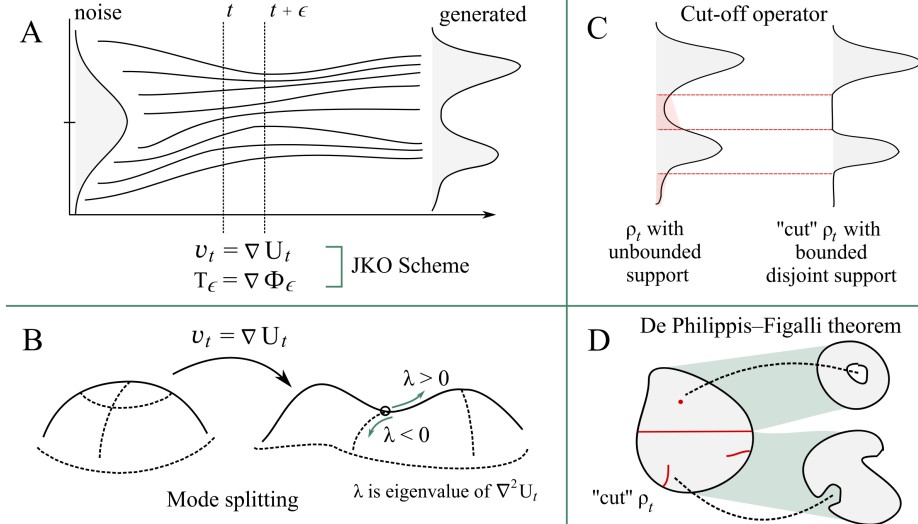

Figure 2: Illustration of our theoretical mechanism behind the formation of hierarchical phase boundaries in diffusion models, that we experimentally observe in Fig 1:
(A) We approximate the reverse probability–flow ODE (PF-ODE) $\dot{x}_t = v(x_t, t)$ by a composition of quadratic–cost optimal transport maps $T_\epsilon = \nabla\Phi_\epsilon$ between successive noisy marginals $\rho_t$ and $\rho_{t+\epsilon}$ using the Jordan–Kinderlehrer–Otto (JKO) scheme.
(B) When the density $\rho_t$ undergoes mode–splitting along the PF-ODE $\dot{x}_t = v(x_t, t) = \nabla U(x_t, t)$, Theorem 1 shows that the heat potential $U$ develops strongly negative curvature, forcing a large local Lipschitz constant of the generative map near the emerging valley between modes.
(C) We apply the cut–off operator $\mathrm{Cut}_\eta$ to the noisy marginals $\rho_t$ to create genuine zero–density gaps between high–density modes, which allows us to use De Philippis-Figalli regularity theorem.
(D) Singularity structure (red lines) of an optimal transportation map $T_\epsilon$ between cut densities $\mathrm{Cut}_\eta[\rho_t]$ and $\mathrm{Cut}_\eta[\rho_{t+\epsilon}]$ derived from De Philippis-Figalli theory.

## 2 THEORETICAL ANALYSIS

### 2.1 NOTATION

We assume all data distributions $\rho$ to be defined on $\mathbb{R}^d$. We denote a $k$-times continuously differentiable function $f : \mathbb{R}^d \to \mathbb{R}$ as $f \in \mathcal{C}^k(\mathbb{R}^d; \mathbb{R})$, and Hölder continuity with exponent $\alpha \in (0, 1]$ of order $k$ as $\mathcal{C}^{k,\alpha}$. Open balls of radius $r > 0$ are denoted $B_r \subset \mathbb{R}^d$, the $d$-dimensional Lebesgue measure is denoted $\mathcal{L}^d$, and $W_2$ denotes the Wasserstein-2 distance. We denote $\nabla := \nabla_{\mathbf{x}}$ the spatial gradient, $\frac{\mathrm{d}}{\mathrm{d}t}$ the time derivative. While all vectors and matrices are in bold (e.g. $\mathbf{x}$ resp. $\mathbf{A}$), we reserve the right to denote the indicator function over the set $A$ with $\mathbf{1}_A$.

### 2.2 DIFFUSION GENERATIVE MODELS

We consider a data distribution $\rho_{\mathrm{data}} \in \mathbb{R}^d$ which we will identify as the starting point of our diffusion process. The noising (forward) process admits a stochastic differential equation formulation (Song et al., 2021)

$$\mathrm{d}\mathbf{x}_t = f(\mathbf{x}_t, t)\mathrm{d}t + g(t)\mathrm{d}\mathbf{B}_t, \quad \mathbf{x}_0 \sim \rho_{\mathrm{data}}. \tag{1}$$

where $B_t$ is a standard Brownian Motion. An equivalent representation at time $t > 0$ is a convolution of $\rho_0$ with the Gaussian kernel $\gamma_t = \mathcal{N}(0, \boldsymbol{\Sigma}_t)$ with $\boldsymbol{\Sigma}_t := t\mathbf{I}_d$, such that $\rho_t := \rho_0 * \gamma_t$.

The forward process turns samples into Gaussian noise exponentially fast (Bakry et al., 2013), allowing the existence of a finite stopping time $T > 0$ such that $\rho_T \approx \mathcal{N}(0, \boldsymbol{\Sigma}_T)$. There also exists a reverse SDE (Anderson, 1982),

$$\mathrm{d}\mathbf{x}_t = \left(f(\mathbf{x}_t, t) - g^2(t)\nabla\log\rho_t(\mathbf{x}_t)\right)\mathrm{d}t + g(t)\mathrm{d}\tilde{\mathbf{B}}_t, \quad \mathbf{x}_T \sim \rho_T, \tag{2}$$

leading us to *score-based generative models*, where $\tilde{\mathbf{B}}_t$ is an another realization of standard Brownian Motion; $t$ is also reversed and changes from $T$ to $0$. The reverse SDE formulation allows for the

*probability flow ODE* (PF-ODE) formulation (Song et al., 2021)

$$d\mathbf{x}_t = \left( f(\mathbf{x}_t) - \frac{1}{2} g^2(t) \nabla \log \rho_t(\mathbf{x}_t) \right) dt \equiv \nabla U(\mathbf{x}_t, t) dt. \tag{3}$$

In the upcoming sections, we will study formation of high-Lipschitz constants and singular sets in the velocity field $v(\mathbf{x}_t, t) = \nabla U(\mathbf{x}_t, t)$, where $U(\mathbf{x}_t, t)$ is the "heat" potential.

Note that $g$ represents noise schedule and is always a smooth function and drift $f$ is linear in $x$ for VP-SDE Song et al. (2021). The Lipschitz constant of equation 3 depends on the gradient of velocity field $\nabla v(\mathbf{x}_t, t) = \nabla^2 U(\mathbf{x}_t, t)$. Therefore, the core source of irregularity lies in the hessian of the log-density $\nabla^2 \log \rho_t(\mathbf{x}_t)$. A more thorough analysis is deferred to Appendix A.1.

## 2.3 MODE SPLITTING LEADS TO A HIGH LOCAL LIPSCHITZ CONSTANT

When considering real image datasets, mode-splitting, which is inevitable during reverse-ODE sampling, produces negative eigenvalues of $\nabla^2 \log \rho_t(\mathbf{x}_t)$ near the "valley" separating two modes (see Fig. 2B). When the most negative eigenvalue becomes sufficiently large in magnitude, the local Lipschitz constant of the reverse ODE becomes large. In other words, the formation of negative curvature during mode-splitting forces a large local Lipschitz modulus for the PF-ODE drift $\nabla U$. Formally, this can be summarized in the following Theorem (whose proof is provided in Appendix A.2).

**Theorem 1.** *Let $\rho \in \mathcal{C}^4(\mathbb{R}^d)$ be a strictly positive probability distribution and fix a compact set $K \subset \mathbb{R}^d$ with open balls $B_r, B_{2r} \subset K$. For $\epsilon > 0$, set $\rho_\epsilon = \rho * \gamma_\epsilon$ and let $\Phi_\epsilon$ be the Brenier potential such that $(\nabla \Phi_\epsilon)_{\#} \rho_\epsilon = \rho$. Define the heat potential as $U(x) = \frac{1}{2} \|\mathbf{x}\|^2 + \epsilon \psi(\mathbf{x})$ with $\psi = -\frac{1}{2} \log \rho$. Assume on $B_{2r}$ that for some $0 < m \leq M < \infty$ and $C_0 < \infty$ we have*

$$0 < m \leq \rho_\epsilon < M, \qquad \text{and} \qquad \|\rho\|_{\mathcal{C}^{2,\alpha}(B_{2r})} \leq C_0. \tag{4}$$

*Then,*

1. *There exists a constant $C \equiv C(d, m, M, C_0) > 0$ such that*

$$\|\nabla^2 \Phi - \nabla^2 U\|_{\mathcal{C}^\alpha(B_r)} \leq C\epsilon^2. \tag{5}$$

2. *If $\nabla^2 U$ undergoes mode splitting, i.e. there exists $\eta > C\epsilon^2 > 0$ such that $\lambda_{\min}(\nabla^2 U) < -\eta$, then the Lipschitz constant of equation 3 is lower-bounded by $\eta$.*

This theorem establishes a large local Lipschitz constant near saddle points of $\log \rho_t(\mathbf{x}_t)$. However, in the Experiment section, we observe not separate points, but rather contiguous regions. To explain their emergence, we refer to the JKO scheme and Caffarelli's regularity theory.

## 2.4 HIERARCHICAL SINGULAR SET ACCUMULATION IN REVERSE PF-ODE

The evolution of density during the solution of the reverse ODE could be described by the Fokker-Planck equation

$$\partial_t \rho_t = \nabla \cdot (\rho_t \nabla V) + \Delta \rho_t. \tag{6}$$

Classical theorem of Jordan, Kinderlehrer and Otto Jordan et al. (1998) states that equation 6 is the $W_2$-gradient flow of the free energy

$$\mathcal{F}(\rho) = \int \rho \log \rho \, d\mathbf{x} + \int_{\mathbb{R}^d} V(\mathbf{x}) \, \rho(d\mathbf{x}). \tag{7}$$

Direct corollary of this theorem is that the solution of the reverse ODE could be approximated with arbitrary accuracy via composition of optimal transport maps for quadratic cost.

Now we want to apply results of the De Philippis-Figalli regularity theorem De Philippis & Figalli (2015), which states that if the target distribution has a non-convex or disjoint support, then the optimal transport map will be discontinuous with a measure-zero singularity set. Moreover, except for a countable number of points, the singular set is differentiable.

The main technical problem that arises here is that the marginal density $\rho_t$ at time $t > 0$ has full support and its density never becomes zero, so the De Philippis-Figalli theorem is not directly applicable. To solve this, we introduce the Cut–off Operator, which sets the density to zero if it is smaller than a cutoff and then renormalizes it ("cut trick").

**Cut–off operator and mode separation**   Given a (noisy) density $\rho_t$ with finite second moment and a threshold $\eta > 0$, define the cut–off operator

$$\mathrm{Cut}_\eta(\rho_t) \; := \; \frac{\rho_t \, \mathbf{1}_{\{\rho_t > \eta\}}}{\int_{\{\rho_t > \eta\}} \rho_t \, dx}, \tag{8}$$

and we write $\widehat{\rho}_t^{\,\eta} := \mathrm{Cut}_\eta(\rho_t)$. For any null set $E$, $\int_E \widehat{\rho}_t^{\,\eta} \, dx = 0$, hence $\widehat{\rho}_t^{\,\eta} \ll \mathcal{L}^n$ and admits a Lebesgue density (see Lemma 7), and an associated quadratic cost Brenier potential thus exists (Corollary 2) such that $\widehat{\Phi}_{t \to t + \epsilon t}^{\eta}$ with $(\nabla \widehat{\Phi}_{t \to t + \epsilon t}^{\eta})_\# \, \widehat{\rho}_t^{\,\eta} = \widehat{\rho}_{t+\epsilon t}^{\,\eta}$, which we shorthand to $\widehat{\Phi}_t^{\eta}$.

By Lemmas 5 resp. 7, we state that the cut measure $\widehat{\rho}_t^{\,\eta}$ is not only close to its uncut counterpart $\rho_t$ in the $W_2$-sense, but the Brenier potentials also converge uniformly on compact sets

$$\widehat{\Phi}_t^{\eta} \to \Phi_t, \quad \text{as } \eta \downarrow 0. \tag{9}$$

Now, we follow the "Union of Manifolds Hypothesis" (Brown et al. (2023)), and assume that the real data distribution is supported on disjoint sets, which gives rise to the following assumption, Fig. 2C:

**Assumption 1.** *There exists $\eta \in (0, \eta_0]$ such that $A_t^\eta := \{\rho_t > \eta\}$ and $A_{t+\epsilon t}^\eta := \{\rho_{t+\epsilon t} > \eta\}$ decompose into a finite union of pairwise separated open sets*

$$A_s^\eta = \bigsqcup_{i=1}^{m_s} \Omega_{s,i}^\eta, \qquad dist(\Omega_{s,i}^\eta, \Omega_{s,j}^\eta) \geq d_\eta > 0 \quad (i \neq j), \tag{10}$$

*with $\widehat{\rho}_s^{\,\eta}$ locally $\mathcal{C}^\alpha$ and two-sided bounded on each component:*

$$0 < c_\eta \leq \widehat{\rho}_s^{\,\eta} \leq C_\eta \quad \text{on each } \Omega_{s,i}^\eta, \qquad s \in \{t, t + \epsilon t\}. \tag{11}$$

*Moreover at least one of $A_t^\eta$ or $A_{t+\epsilon t}^\eta$ is* non-convex.

**Singular sets between modes**   Due to the application of the cut-off operator in Eq 8, we obtain densities that satisfy the assumptions of the De Philippis–Figalli theorem 1.3 (De Philippis & Figalli, 2015). Based on it, we now state that $\widehat{\Phi}_t^{\eta}$ is strictly convex on each component $\Omega_{t,i}^\eta$ and due to assumed disjoint support of the target distribution, it will necessarily form *singular sets*. We denote such sets $\Sigma_\eta \subset A_t^\eta$ and depict them as red lines on Fig. 2D. In Theorem 2 we show that $\widehat{\Phi}_t^{\eta}$ is differentiable on $A_t^\eta \backslash \Sigma_\eta$, $\widehat{T}_\eta$ is continuous on $A_t^\eta \backslash \Sigma_\eta$, the singular set $\Sigma_\eta$ has Lebesgue measure zero, and that $\widehat{\Phi}_t^{\eta}$ develops a *kink (a point of non-differentiability)* in a neighbourhood of $\Sigma_\eta$. All these statements are the direct corollary of the De Philippis-Figalli theorem.

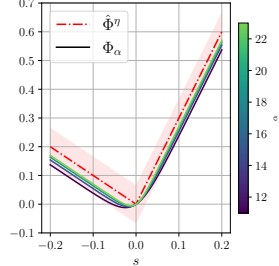

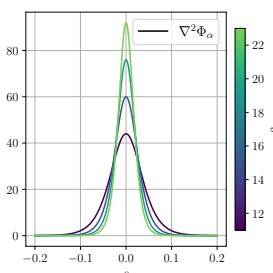

Figure 3: Local uniform closeness of smooth approximants $\{\Phi_\alpha\}_\alpha$ undergo curvature blowup near the kink.

Now, after studying the singularities for the the cut measures $\widehat{\rho}_t^{\,\eta}$, we return to the original densities $\rho_t$ and relate the singular structure of optimal transports maps $\widehat{T}_\eta = \nabla \widehat{\Phi}_t^{\eta}$ to that of $T = \nabla \Phi_t$. Since the cut and uncut densities are close in the Wasserstein distance, the cut Brenier potentials $\widehat{\Phi}_t^{\eta}$ converge uniformly to original $\Phi_t$ as $\eta \to 0$. We will use this closeness to show that non-Lipschitz boundaries for the cut measures will turn into high Lipschitz regions in the original case. We achieve it using the following kink analysis.

**Kink analysis and curvature growth**   We will assume that there exists $r > 0$ and a unit vector $\mathbf{v} \in \mathbb{R}^d$, which is normal to the singular set $\Sigma_\eta$, such that the one-dimensional restriction

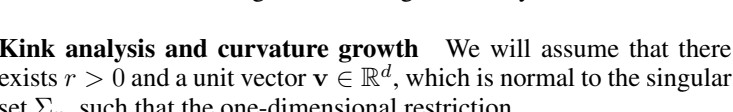

$$g_\eta(s) := \widehat{\Phi}_t^{\eta}(\mathbf{x}_0 + s\mathbf{v}), \quad s \in (-r, r), \quad \mathbf{x}_0 \in \Sigma_\eta \tag{12}$$

is convex, but has a derivative jump $\Delta_\eta := g_{\eta,+}'(0)$. We show that if the cutoff potential $\widehat{\Phi}^\eta$ exhibits a jump in derivatives along at least a one-dimensional direction $\mathbf{v} \in \mathbb{R}^d$, then $\|\nabla^2 \Phi\|_{\mathrm{op}}$, and therefore the Lipschitz constant of the reverse probability flow ODE, has a strictly positive lower-bound (see Theorem 3).

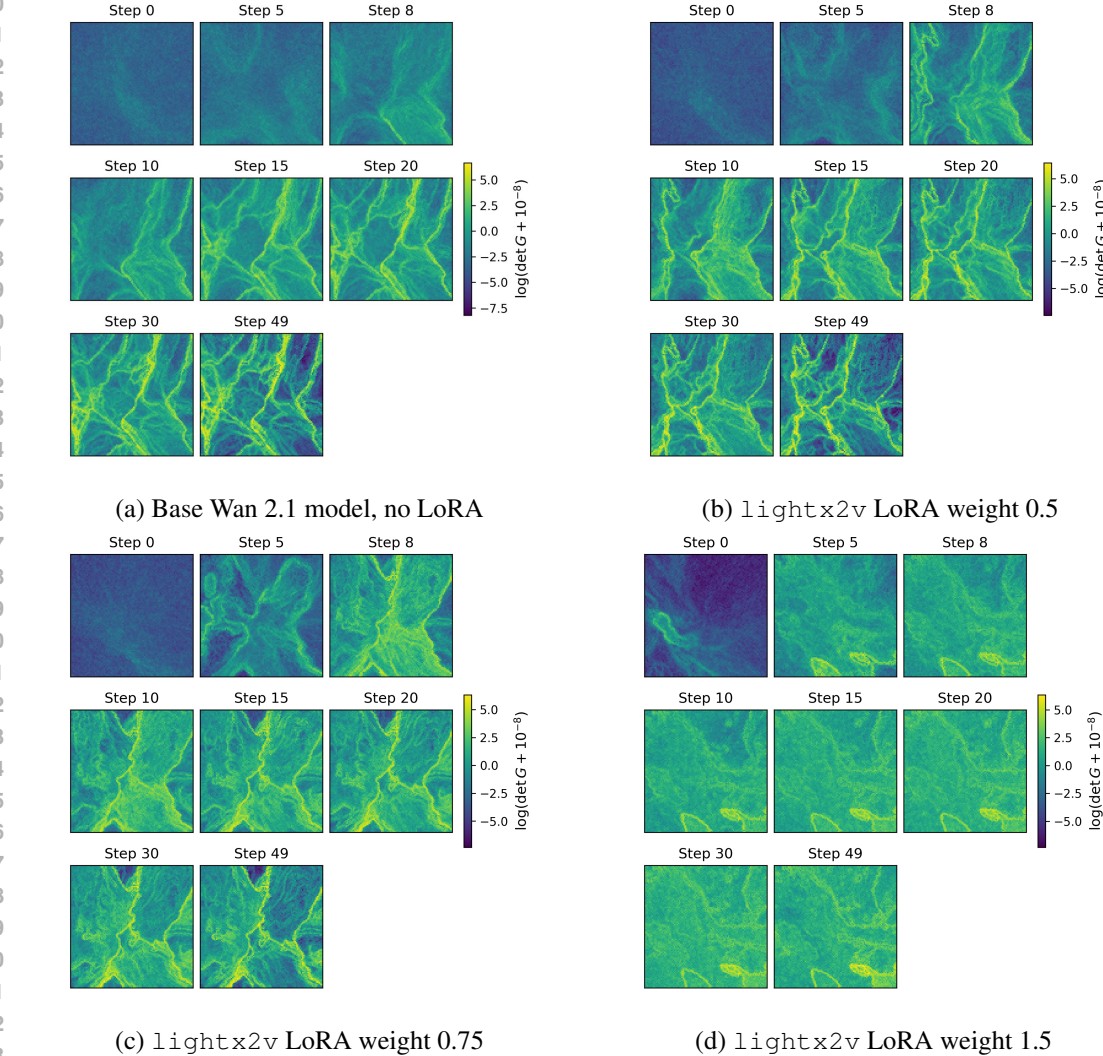

(a) Base Wan 2.1 model, no LoRA

(b) `lightx2v` LoRA weight 0.5

(c) `lightx2v` LoRA weight 0.75

(d) `lightx2v` LoRA weight 1.5

Figure 4: Heatmaps of metric determinant $\sqrt{\det G}$ in a 50-step diffusion process. Baseline Wan 2.1 is compared against generations with varying weights of acceleration LoRA. Panels: (a) No LoRA; (b) LoRA 0.5; (c) LoRA 0.75; (d) LoRA 1.5. Bright ridges indicate high-Lipschitz boundaries.

**Example 1.** *We demonstrate this phenomenon with a simple example: $\widehat{\Phi}^{\eta}(s) = \max\{-s, 3s\}$ clearly develops a kink around $s = 0$, which we smoothly approximate via parametric log-sum-exp $\Phi_{\alpha}(s) := \alpha^{-1}\left(e^{-\alpha s} + e^{3\alpha s} - \log(2)\right)$. We see that as $\alpha$ increases, the curvature drastically sharpens, depicted in Figure 3.*

**Hierarchical singular set accumulation**   We now finally transfer all the results from above to the PF-ODE from equation 3. As mentioned in the introduction, we approximate our reverse dynamics by a JKO scheme: a composition of convex function gradients over small time-intervals. The nature of the JKO scheme implies that high-Lipschitz boundaries and singular sets $\Sigma_{\eta}$ will *accumulate and propagate* through the PF-ODE as we progress through the generative process.

## 3   METHODS

**Phase boundaries evolution** We investigate the formation of phase boundaries in the latent space of Stable Diffusion 1.5 (Dreamshaper8 by Lykon (2023)) and Wan 2.1 text-to-video model Wan-AI Team (2025), sampling one frame (text-to-image regime). To study the Lipschitz constant of the generative map, we use a bound that relates it to the determinant of the pullback metric (see

| Model | Edge count | Avg. sharpness | CLIP Diversity↑ | SigLIP Diversity↑ |
|---|---|---|---|---|
| *50 steps* | | | | |
| base model | **5502** | **1462.3** | **0.4151±0.0007** | **0.3411±0.0005** |
| lightx2v, $w = 0.50$ | 5451 | 1388.5 | 0.3729±0.0007 | 0.3144±0.0006 |
| lightx2v, $w = 0.75$ | 3897 | 1038.0 | 0.2607±0.0006 | 0.2469±0.0005 |
| lightx2v, $w = 1.50$ | 1045 | 562.6 | 0.0989±0.0004 | 0.1103±0.0004 |
| *5 steps* | | | | |
| lightx2v, $w = 1.0$ | 3851 | 1025.7 | 0.3323±0.0007 | 0.3114±0.0006 |
| CausVid, $w = 1.0$ | 2317 | 794.2 | 0.1970±0.0005 | 0.2064±0.0006 |

Table 1: Wan 2.1. Numerical evaluation of Fig. 4. Dataset statistics for different acceleration LoRA weights. Edge count is the number of pixels detected as edges by the Canny detector on the resized $150 \times 150$ grayscale image. Average sharpness is the mean gradient magnitude (computed via $5 \times 5$ Sobel filters) over the whole grid. Image diversity is computed as the average cosine distance between CLIP or SigLIP embeddings of $5 \times 10^4$ randomly sampled image pairs from the slice.

Lemma 9).

$$\text{Lip}(F; z) \geq \left( \sqrt{\det G(z)} \right)^{1/2}. \tag{13}$$

This bound implies that if we observe a large determinant of the pullback metric in latent space, then the generative map has a large Lipschitz constant. We use CLIP embeddings of generated images at different noise levels to estimate the metric tensor on a 2D latent slice, following the data generation procedure by Lobashev et al. (2025).

We fix three latent anchors $z_0, z_1, z_2 \in \mathbb{R}^k$ and parametrize a two-dimensional affine slice by

$$z'(\alpha, \beta) = z_0 + \alpha (z_1 - z_0) + \beta (z_2 - z_0), \qquad z(\alpha, \beta) = z'(\alpha, \beta)/||z'(\alpha, \beta)||. \tag{14}$$

Starting from the latent noises from this slice $z(\alpha, \beta)$, we run the diffusion sampler with a generic prompt "High quality picture, 4k, detailed" to obtain the predicted clean images $\hat{x}_0(\alpha, \beta; t)$ (Fig. 1, leftmost column). Then we embed images with CLIP (ViT-B/32) (Radford et al., 2021) to get a feature $f(\alpha, \beta; t) = \phi(\hat{x}_0(\alpha, \beta; t)) \in \mathbb{R}^d$. On a uniform $N \times N$ grid of initial noises with $\alpha_i = i/(N-1)$, $\beta_j = j/(N-1)$, we estimate the Jacobian of the feature map $f$ by centered finite differences and one–sided differences at the grid boundary. Stacking the columns gives $J(\alpha, \beta; t) \in \mathbb{R}^{d \times 2}$, and the pullback metric on the slice is

$$G(\alpha, \beta; t) = J(\alpha, \beta; t)^\top J(\alpha, \beta; t) \in \mathbb{R}^{2 \times 2}, \qquad (\alpha, \beta) \in [0, 1]^2. \tag{15}$$

For each $t$, we render heatmaps of $S(\alpha, \beta; t) = \sqrt{\det G}$ on $[0, 1]^2$. $S$ has the meaning of a local area expansion factor. For visualization we display $S$ in log scale, see Fig. 1 and Fig. 8. We observe that: (i) for high-noise regime, $S \approx$ const (flat metric); (ii) as $t$ decreases, narrow high–$S$ ridges appear and branch, partitioning the slice into phases; (iii) in the low-noise regime, the ridge pattern stabilizes and the mean of $S$ approaches a plateau (see Fig. 16 and steps 19-39 in Fig. 1).

**Effect of distillation** We study the effect of Distribution Matching Distillation (DMD) on Wan 2.1 text-to-video model. We compare the base model against models with acceleration LoRAs `lightx2v` (Kijai, 2025b) (DMD2/Self-Forcing distillation) and `CausVid` (Kijai, 2025a) (DMD/Autoregressive distillation), using the same generic prompt. This approach is particularly beneficial because acceleration LoRAs allow us to use the same architecture, sampler, and initial latent noises as for the base model. The heatmap of $S$ for `lightx2v` under varying LoRA strength is given in Fig. 4. Empirically, we observe that the acceleration LoRA shifts phase boundaries toward earlier generation steps. Table 1 shows quantitatively that both the sharpness and the number of boundaries decrease when the acceleration LoRA is applied. At the same time, the diversity of generated images is reduced, indicating worse mode coverage. This statement is valid for 50 steps and for 5-step accelerated schedule, Table 1.

**Robustness across feature extractors** Ridge locations in the final-step $\sqrt{\det G}$ maps are consistent across CLIP, DINOv2, SigLIP, and pixel ($\ell_2$), indicating that phase boundaries reflect properties of the generative map rather than a specific embedding, see Fig. 14.

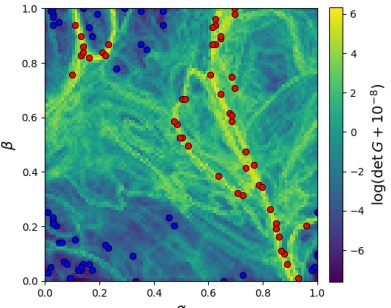

Figure 5: Sampling locations (red: edges with high $\det(G)$, blue: basins with low $\det(G)$). Reconstruction metrics for these locations are summarized in Table 2.

| Metric (mean±std) | Boundary | Basin |
|---|---|---|
| MSE↓ | $0.035 \pm 0.001$ | $0.020 \pm 0.001$ |
| SSIM↑ | $0.618 \pm 0.004$ | $0.740 \pm 0.003$ |
| CLIP↑ | $0.711 \pm 0.008$ | $0.839 \pm 0.007$ |
| DINOv2↑ | $0.397 \pm 0.015$ | $0.683 \pm 0.014$ |

Table 2: Reconstruction metrics for DDIM inversion (mean ± std) for boundary vs. basin points; see sampling locations in Fig. 5. The DDIM inversion reconstruction errors for images located near the boundaries are consistently higher than those for images far from the boundaries inside basins, across all metrics.

**Effect of classifier guidance** In our experiments, classifier-free guidance (CFG) increases the average geometric sensitivity of the slice: the feature-space entropy rise relative to the baseline, see Fig. 15. The impact of high CFG on boundary formation reminds one of acceleration LoRAs, but the shift toward earlier steps is less prominent, Fig. 17.

The presented experiments cover two models, that have different size, architecture and training procedure. This, together with our theoretical analysis, lead us to the idea that high-Lipschitzness and hierarchical appearance of boundaries are general properties of the unimodal-to-multimodal diffusion mappings. To demonstrate the emergence of the hierarchical phase boundaries from multimodal data with disjoint support, we provide the following 1D example.

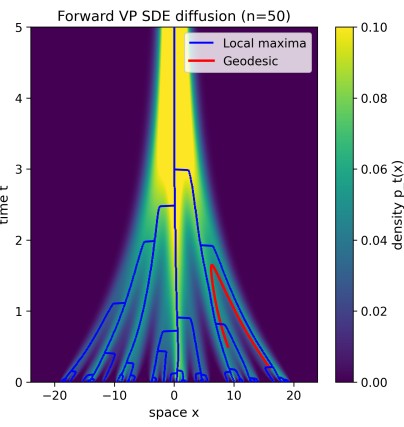

Figure 6: Spacetime density heatmap with maxima (blue) and geodesic (red, calculated following Karczewski et al. (2025))

**1D Diffusion** We consider the 1D variance-preserving (VP) SDE. Let the initial data be an empirical distribution with atoms $\{x_{0,i}\}_{i=1}^n$ sampled uniformly on $[-a, a]$ with $a = 20$. The forward marginal at time $t$ is the equal-weight Gaussian mixture

$$p_t(x) = \frac{1}{n} \sum_{i=1}^n \mathcal{N}\big(x;\ \alpha(t)\, x_{0,i},\ \sigma^2(t)\big). \qquad (16)$$

We evaluate $p_t(x)$ on a spacetime grid to produce a density heatmap using the exact solution Eq. 16.

For each time slice $x \mapsto p_t(x)$, we detect local maxima using a relative-height threshold that adapts to the slice's scale. This yields a piecewise-linear "mode evolution tree" (blue line segments), revealing branching as diffusion progresses, see Figure 6. Interestingly, one may consider a distance between samples as a distance on the spacetime tree. Moreover, in this 1D example tree distance resembles a spacetime geodesic length by Karczewski et al. (2025), see Fig. 6. The distance defined by tree is following the strong triangle inequality and thus is an example of the ultrametric distance. For a discussion of ultrametricity in higher dimensions, see Proposition 3.

**DDIM inversion and image editing** Having images in one phase before a certain time means that their latents share features and their noiseless predictions point to similar images, Fig. 7 and Fig. 12. Reversing a given image back in time via the PF-ODE is the core idea of DDIM inversion, a well-known technique for image editing (Mokady et al., 2023). High-Lipschitz constant of diffusion mapping means an instability of PF-ODE solutions, therefore affecting the numerical stability of the DDIM inversion. We show this quantitatively by measuring the DDIM inversion reconstruction error for noise samples near high-Lipschitz boundaries versus noise samples located at low-Lipschitz,

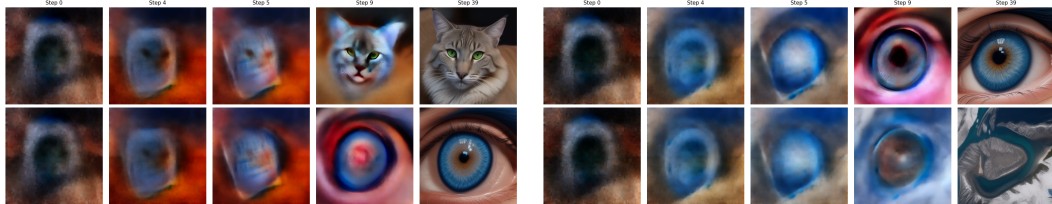

Figure 7: Due to high Lipschitz constant of PF-ODE, close latent noises yield similar noiseless predictions before crossing a phase boundary; after crossing, they diverge.

regular basins, Fig.5. Table 2 demonstrate that the DDIM inversion reconstruction errors for images located near the boundaries are consistently higher than those for images far from the boundaries inside basins, across all metrics. This experiment clearly demonstrates the correspondence between the computed pullback metric and the Lipschitz constant of the diffusion mapping.

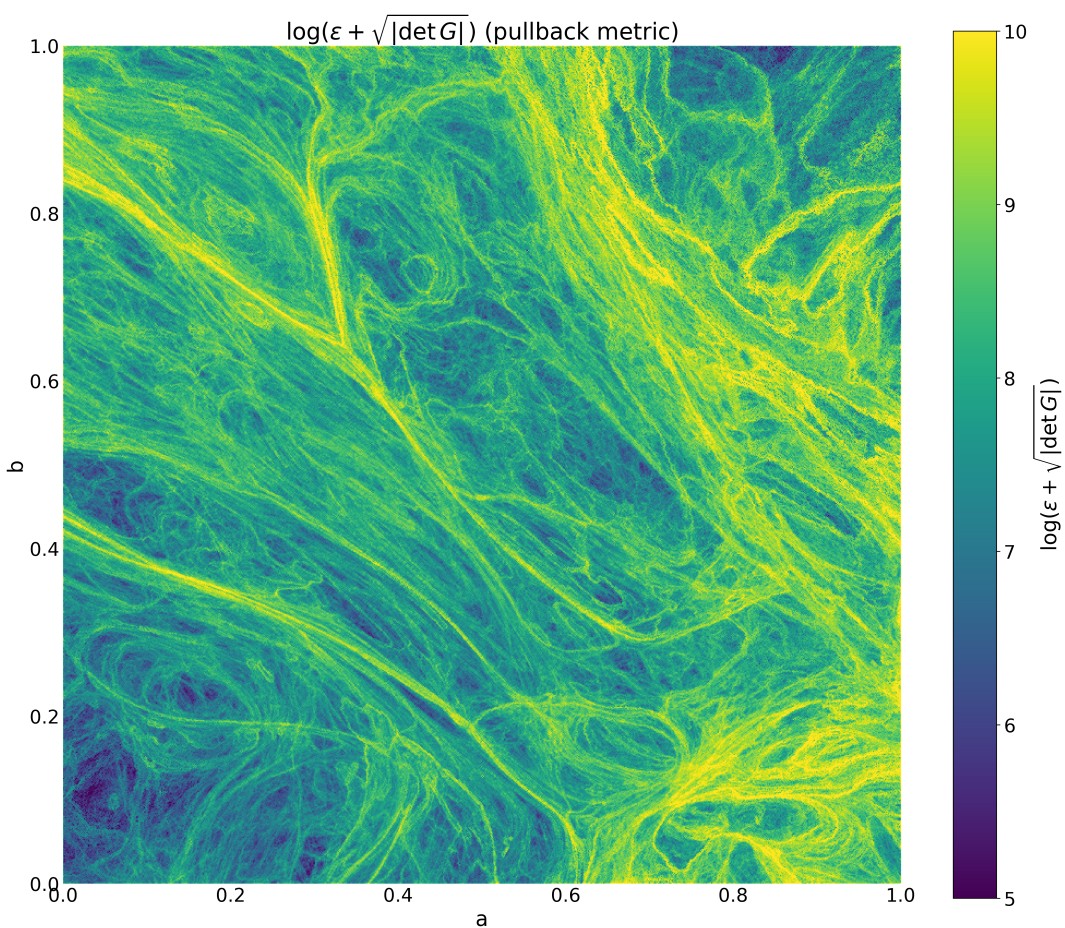

Figure 8: A high-resolution visualization of the pullback metric determinant on a 2D slice of the latent space for SD- 1.5 Dreamshaper (prompt: "Painting"). The heatmap shows $\log(\varepsilon + \sqrt{|\det G|})$, where larger values indicate higher sensitivity of the generated image CLIP embeddings to perturbations in $(a, b)$. The grid resolution is $768 \times 768$.

## 4 RELATED WORK

Lei et al. Lei et al. (2019) apply Caffarelli's regularity theory to analyze mode collapse in GANs. In contrast, we apply Caffarelli's theory to diffusion models and combine it with the Jordan-Kinderlehrer-Otto (JKO) scheme for the Fokker-Planck equation. Similar observations of sharp changes in diffusion models were reported in Humayun et al. (2025); however, we analyze phase boundaries at intermediate reverse-time steps and study the effects of diffusion distillation and classifier-free guidance. Closely related, Lobashev et al. (2025) computes an approximate Fisher information metric for diffusion models and observes phase boundaries at the final timestep; we complement this by tracking their hierarchical formation across time.

## 5 DISCUSSION

In this work, we study the regularity of diffusion mapping via the pullback metric on a latent space. First, we establish a relation between the pullback metric and the Lipschitz constant and associate high curvature of a pullback metric with a phase boundary. Then, on a two-dimensional latent slice, we experimentally observe a hierarchical accumulation of phase boundaries as the diffusion process progresses. To explain the mechanism behind this emergence we refer to the JKO scheme and regularity theory of optimal transport maps. We theoretically show that the observed high-Lipschitz boundaries are inherent properties of any unimodal-to-multimodal diffusion mappings. To further prove this idea, we study text-to-image (SD1.5) and text-to-video (Wan 2.1) models, and demonstrate that despite a different size, domain and architecture, these models exhibit a similar behavior in term of boundary formation.

We additionally examine how phase formation changes under different settings: DMD distillation LoRAs and varying classifier-free guidance. Empirically, they shift a formation of boundaries towards earlier generation steps, but decrease their average number and sharpness. This correlates with degraded diversity of samples. In other words, few-step generation comes with a price of lower Lipschitz constant and worse mode coverage. This leads to an important practical implication: non-Lipschitzness of generative mapping is necessary for good mode coverage. However, non-Lipschitzness is often seen as an undesirable property. For instance, earlier works Guo et al. (2024); Hahm et al. (2024); Zhou et al. (2025) smooth the latent space, effectively reducing the spatial Lipschitz constant. Our findings suggest that these constraints will suppress model expressive power. It also suggests that many-step generation may be necessary to develop a high enough Lipschitz constant, at least with the current model design. At the same time, non-Lipschitzness of the diffusion model may cause training instabilities and possible gradient explosions. The presented paper, therefore, opens a discussion on balancing these two aspects.

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

# A    PROOFS FROM SECTION 2

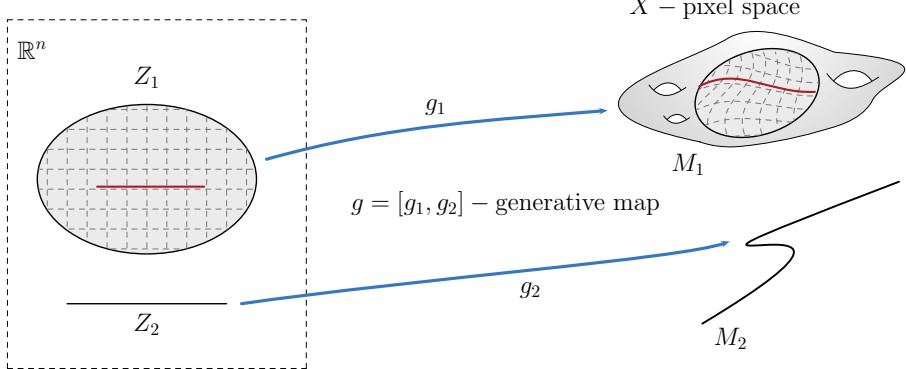

Figure 9: Differential-geometric view on the generative modeling problem: observed data $x \in X$ concentrate near a low-dimensional manifold $M \subset X$ generated by a map $g : Z \to M$ from latent variables $z \in Z \subset \mathbb{R}^n$.

## A.1    LOCAL LIPSCHITZ ANALYSIS

We assume standard well-posedness for the forward SDE 1: $f(\cdot, t)$ is globally Lipschitz with at most linear growth, and the diffusion is non-degenerate (Song et al., 2021; Oksendal, 2013). Clearly, the (learned) score function $\nabla \log \rho_t \propto \nabla U(\cdot, t)$ in equation 3 is what will determine unusual dynamics in the generation process. We now present a statement on the driving element Hessian of the heat potential.

**Proposition 1.** *Let $t > 0$ and consider $\rho_t$ as above. Assume $\nabla \rho_t(\mathbf{x}_\star) = 0$. Then*

$$\nabla^2 \log \rho_t(\mathbf{x}_\star) \;=\; \frac{\nabla^2 \rho_t(\mathbf{x}_\star)}{\rho_t(\mathbf{x}_\star)}. \tag{17}$$

*In particular, for any matrix norm $\| \cdot \|$, we have $\left\| \nabla^2 \log \rho_t(\mathbf{x}_\star) \right\| \;=\; \frac{\left\| \nabla^2 \rho_t(\mathbf{x}_\star) \right\|}{\rho_t(\mathbf{x}_\star)}$.*

As a consequence of Proposition 1, we have that low-probability stationary points of $\rho_t$ influence the Lipschitz constant of the PF-ODE 3: as the distribution approaches the low-noise collapse phase of Biroli et al. (2024), the Lipschitz constant gets amplified due to the emergence of zero-probability regions. We will discuss in more detail how this growth may lead to divergence in Section 2.4.

*Proof.* By the chain rule, we have

$$\nabla \left( \nabla \log \rho_t(\mathbf{x}_\star) \right) = \nabla \left( \frac{\nabla \rho_t(\mathbf{x}_\star)}{\rho_t(\mathbf{x}_\star)} \right)$$
$$= \frac{\nabla^2 \rho_t(\mathbf{x}_\star)}{\rho_t(\mathbf{x}_\star)} - \frac{(\nabla \rho_t(\mathbf{x}_\star))(\nabla \rho_t(\mathbf{x}_\star))^T}{(\nabla \rho_t(\mathbf{x}_\star))^2}, \tag{18}$$

Evaluating at the stationary point $\mathbf{x}_\star$ implies vanishing of the second term. $\square$

## A.2    PROOF OF THEOREM 1

We proceed by proving the Theorem in three steps.

### BOUNDING THE HESSIAN DIFFERENCE

We denote the Brenier potential as

$$\Phi(\mathbf{x}, t) = \frac{1}{2} \|\mathbf{x}\|^2 + \epsilon \psi_\epsilon, \tag{19}$$

and the heat potential by

$$U(\mathbf{x}, t) = \frac{1}{2}\|\mathbf{x}\|^2 + \epsilon\psi, \quad \psi = -\frac{1}{2}\log\rho. \tag{20}$$

Taking the $\mathcal{C}^\alpha$-norm of their difference, we obtain

$$\|\nabla^2\Phi - \nabla^2 U\|_{\mathcal{C}^\alpha(B_r)} = \epsilon\|\nabla^2\phi\|_{\mathcal{C}^\alpha(B_r)}, \tag{21}$$

with $\phi := \psi_\epsilon - \psi$. Recalling the Hölder norm definition, we have $\|\nabla^2\phi\|_{\mathcal{C}^\alpha(B_r)} \leq \|\phi\|_{\mathcal{C}^{2,\alpha}(B_r)}$, allowing upper-bounding of equation 21 via the norm of $\phi$. Using Lemma 2 we obtain

$$\|\nabla^2\Phi - \nabla^2 U\|_{\mathcal{C}^\alpha(B_r)} \leq C'\epsilon^2 \tag{22}$$

WEYL BARRIER

Using the definition $\|\nabla^2\phi\|_{\mathcal{C}^\alpha} = \|\nabla^2\phi\|_{L^\infty} + [\nabla^2\phi]_{\mathcal{C}^\alpha}$, equation 22 implies boundedness of $\nabla^2\phi$ in the sense of the $L^\infty$-norm. This in turn implies boundedness of the operator bound[1]

$$\sup_{B_r}\|\nabla^2\Phi - \nabla^2\mathcal{U}\|_{op} \leq C\epsilon^2. \tag{23}$$

We next recall Weyl's inequality for two Hermitian operators $A, B$, with ordering of the eigenvalues as $\lambda_1 \geq \cdots \geq \lambda_n$:

$$|\lambda_k(A + B) - \lambda_k(A)| \leq \|B\|_{op}. \tag{24}$$

Plugging $A = \nabla^2 U$ and $B = \nabla^2\phi := \nabla^2\Phi - \nabla^2 U$ into equation 24 gives us the bound

$$|\lambda_k(\nabla^2\Phi) - \lambda_k(\nabla^2 U)| \leq \|\nabla^2\phi\|_{op}. \tag{25}$$

The above provides control over the minimal eigenvalue, for which we know $\lambda_{\min}(\nabla^2\Phi) \geq 0$ by convexity of the Brenier potential. Combining the spectral control with the bound on the operator norm of $\nabla^2\phi$ gives us

$$\lambda_{\min}(\nabla^2 U) \geq -C\epsilon^2, \quad \text{on } B_r. \tag{26}$$

The second statement is a direct consequence of Weyl's inequality.

□

### A.3 SINGULAR SET FORMATION: CAFFARELLI/FIGALLI-DE PHILIPPIS INTERIOR REGULARITY

**Proposition 2** (Interior regularity on each mode). *On each component $\Omega_{t,i}^\eta$ the potential $\widehat{\Phi}^\eta$ is strictly convex and $C^{1,\alpha}$ in the interior (De Philippis & Figalli, 2015), with $D^2\widehat{\Phi}^\eta \in L_{\mathrm{loc}}^{1+\epsilon}(\Omega_{t,i}^\eta)$ (De Philippis & Figalli, 2013).*

**Theorem 2** (Existence of a measure–zero singular interface for the cut pair). *Under Assumption 1, there exists a Borel set $\Sigma_\eta \subset A_t^\eta$ such that:*

1. *$\widehat{\Phi}^\eta$ is differentiable on $A_t^\eta \setminus \Sigma_\eta$ and $\widehat{T}_\eta = \nabla\widehat{\Phi}^\eta$ is continuous there (indeed $\mathcal{C}^{0,\alpha}$ on compact subsets of each $\Omega_{t,i}^\eta$);*

2. *$\Sigma_\eta$ is contained in the set where the supremum defining $\widehat{\Phi}^\eta$ has at least two maximizers;*

3. *$\Sigma_\eta$ has Lebesgue measure $0$ and locally finite $\mathcal{H}^{n-1}$–measure;*

4. *On each connected component of $(A_t^\eta \setminus \Sigma_\eta)$, $\widehat{T}_\eta$ is single–valued and smooth, while $\widehat{T}_\eta$ has distinct one–sided limits when crossing $\Sigma_\eta$ from regions mapped to different target components.*

---

[1]The constant depends on how we take $L^\infty$, max-entrywise or Frobenius. If we consider the entrywise-max norm, we have to absorb $\sqrt{d}$ into the constant $C$.

*Proof.* Items (1) and (3) are consequences of Proposition 2 and the $W^{2,1+\epsilon}$ integrability: convex potentials are $\mathcal{C}^{1,\alpha}$ off of the singular set, which has measure zero. Item (2) follows from the Fenchel representation:

$$\widehat{\Phi}^\eta(\mathbf{x}) = \sup_{\mathbf{y} \in A^\eta_{t+\epsilon t}} \{\langle \mathbf{x}, \mathbf{y} \rangle - \widehat{\psi}_\eta(\mathbf{y})\}.$$

Non-differentiability occurs precisely where multiple $\mathbf{y}$'s are active. Item (4) follows from cyclical monotonicity: the pre-images of different target components are separated by the tie–set $\Sigma_\eta$, across which the gradient jumps (or, at least, cannot be extended continuously). $\square$

## A.4 FORMAL STATEMENT & PROOF OF SECTION 2.4

**Theorem 3** (High–Lipschitz interfaces for the uncut pair). *Under Assumption 1, for every $\eta$ small enough there exists a neighbourhood $U_\eta$ of $\Sigma_\eta$ such that the Brenier map $T = \nabla\Phi : \rho_t \mapsto \rho_{t+\epsilon t}$ satisfies*

$$\sup_{U_\eta} \|\nabla T\| \geq c' \, d_\eta. \tag{27}$$

*In particular, if the separation $d_\eta$ does not vanish as $\eta \downarrow 0$, then $\sup_{U_\eta} \|DT\| \to +\infty$ along any sequence where the cut valley is restored (i.e. $\eta \downarrow 0$).*

*Proof.* Combine Lemma 5, Lemma 6, Theorem 2, and Lemma 8 as above, noting that $d_\eta$ is a lower bound on the jump of the one–sided gradients across $\Sigma_\eta$ in the cut problem. Any locally uniform approximation of a kink by a $\mathcal{C}^1$ convex function must develop a large slope near the kink; this yields equation 27. $\square$

## A.5 LIPSCHITZ TRANSPORT MAPS FROM CONVEX POTENTIALS

In this subsection we study when a Lipschitz optimal transport map exists. The classical contraction theorem of Caffarelli Caffarelli (1992) guarantees that the optimal transport map for the quadratic cost is Lipschitz provided both the source and target measures are strongly log–concave. However, real data distributions are typically multimodal and do not satisfy this assumption.

A simple way to construct pairs of measures with a Lipschitz OT map beyond the strongly log–concave setting is as follows. Starting from a given (possibly multimodal) source distribution, apply the gradient of a strongly convex function to obtain a second distribution. By Brenier's theorem, the optimal transport map between these two measures is precisely this gradient, and hence is globally Lipschitz. This gives a concrete answer to the question of when there exists a Lipschitz optimal transport map between multimodal distributions. We formalize this in the following theorem.

**Theorem 4.** *Fix $0 < m \leq M < \infty$. Let $\Phi : \mathbb{R}^n \to \mathbb{R}$ be a $\mathcal{C}^2$ convex function whose Hessian satisfies the uniform bounds*

$$m\mathbf{I}_n \leq \nabla^2\Phi(\mathbf{x}) \leq M\mathbf{I}_n \quad \text{for all } \mathbf{x} \in \mathbb{R}^n \tag{28}$$

*in the Loewner order. Set $T := \nabla\Phi$ and, for an arbitrary Borel probability measure $\mu$ on $\mathbb{R}^n$, define the pushforward measure $\nu := T_\#\mu$. Then the following hold.*

1. *(**Bi-Lipschitz and strong monotonicity**) For all $\mathbf{x}, \mathbf{y} \in \mathbb{R}^n$,*

$$m \|\mathbf{x} - \mathbf{y}\| \leq \|T(\mathbf{x}) - T(\mathbf{y})\| \leq M \|\mathbf{x} - \mathbf{y}\| \tag{29}$$

*and*

$$\big(T(\mathbf{x}) - T(\mathbf{y})\big) \cdot (\mathbf{x} - \mathbf{y}) \geq m \|\mathbf{x} - \mathbf{y}\|^2. \tag{30}$$

*In particular, $T$ is injective, $T(\mathbb{R}^n)$ is open and convex, and $T^{-1}$ exists on $T(\mathbb{R}^n)$ with*

$$\|T^{-1}(\mathbf{u}) - T^{-1}(\mathbf{v})\| \leq \frac{1}{m} \|\mathbf{u} - \mathbf{v}\| \quad \forall \mathbf{u}, \mathbf{v} \in T(\mathbb{R}^n). \tag{31}$$

2. *(**Optimality of the gradient map for the quadratic cost**) Let $c(x, y) = \frac{1}{2}\|x - y\|^2$. Among all measurable maps $S : \mathbb{R}^n \to \mathbb{R}^n$ with $S_\#\mu = \nu$, the map $T = \nabla\varphi$ minimizes the Monge cost*

$$\int_{\mathbb{R}^n} \frac{1}{2}\|\mathbf{x} - S(\mathbf{x})\|^2 \, d\mu(\mathbf{x}).$$

  *Equivalently, the coupling $\pi := (\mathrm{Id}, T)_{\#}\mu$ is an optimal plan in $\Pi(\mu, \nu)$ for the Kantorovich problem with cost $c$.*

  3. **(Lipschitz Brenier map when $\mu$ is absolutely continuous)** *If $\mu \ll \mathcal{L}^n$, then $T$ coincides $\mu$-a.e. with the unique Brenier optimal transport map from $\mu$ to $\nu$ for the cost $c$, and in particular the quadratic OT map $\mu \to \nu$ is $M$-Lipschitz.*

*Proof.* We prove each statement separately.

1. For any $\mathbf{x}, \mathbf{y} \in \mathbb{R}^n$, by the fundamental theorem of calculus along the segment $\mathbf{y} + t(\mathbf{x} - \mathbf{y})$ with $t \in [0, 1]$,

$$T(\mathbf{x}) - T(\mathbf{y}) = \int_0^1 \nabla^2 \Phi\big(\mathbf{y} + t(\mathbf{x} - \mathbf{y})\big)(\mathbf{x} - \mathbf{y})\, dt.$$

Taking operator norms and using $\|\nabla^2 \Phi(\cdot)\|_{\mathrm{op}} \le M$ gives the upper bound in equation 29. Taking the inner product with $(x - y)$ and using equation 28,

$$\big(T(\mathbf{x}) - T(\mathbf{y})\big) \cdot (\mathbf{x} - \mathbf{y}) = \int_0^1 (\mathbf{x} - \mathbf{y}) \cdot \nabla^2 \Phi(\cdot)(\mathbf{x} - \mathbf{y})\, dt \ge m\, \|\mathbf{x} - \mathbf{y}\|^2,$$

which is equation 30. By Cauchy–Schwarz,

$$\|T(\mathbf{x}) - T(\mathbf{y})\|\, \|\mathbf{x} - \mathbf{y}\| \ge \big(T(\mathbf{x}) - T(\mathbf{y})\big) \cdot (\mathbf{x} - \mathbf{y}) \ge m\, \|\mathbf{x} - \mathbf{y}\|^2,$$

so the lower bound in equation 29 follows. In particular, $T$ is injective and (by the inverse function theorem plus equation 28) a global $\mathcal{C}^1$ diffeomorphism between $\mathbb{R}^n$ and the open convex set $T(\mathbb{R}^n)$; equation 31 is the standard consequence of strong monotonicity: for $\mathbf{u} = T(\mathbf{x})$, $\mathbf{v} = T(\mathbf{y})$,

$$m\, \|\mathbf{x} - \mathbf{y}\|^2 \le (\mathbf{u} - \mathbf{v}) \cdot (\mathbf{x} - \mathbf{y}) \le \|\mathbf{u} - \mathbf{v}\|\, \|\mathbf{x} - \mathbf{y}\|.$$

2. We begin by establishing a bound on $\int \mathbf{x} \cdot \mathbf{y}\, d\pi$ via the Legendre conjugate of the Brenier potential

$$\Phi^*(\mathbf{y}) := \sup_{\mathbf{x} \in \mathbb{R}^n} \{\mathbf{x} \cdot \mathbf{y} - \Phi(\mathbf{x})\}. \tag{32}$$

By definition we have

$$\mathbf{x} \cdot \mathbf{y} \le \Phi(\mathbf{x}) + \Phi^*(\mathbf{y}), \tag{33}$$

where is equality is attained when $\mathbf{y} = \nabla\Phi(\mathbf{x})$ by first-order conditions. Consequently, for any coupling $\pi \in \Pi(\mu, \nu)$,

$$\int_{\mathbb{R}^n \times \mathbb{R}^n} \mathbf{x} \cdot \mathbf{y}\, d\pi(\mathbf{x}, \mathbf{y}) \le \int_{\mathbb{R}^n} \Phi\, d\mu + \int_{\mathbb{R}^n} \Phi^*\, d\nu, \tag{34}$$

and equality holds for $\pi = (\mathbf{I}_d, \nabla\Phi)_{\#}\mu$ since equation 33 is pointwise an equality on the graph of $\nabla\varphi$. Thus, we can lower-bound the quadratic cost

$$\int \tfrac{1}{2}\|\mathbf{x} - \mathbf{y}\|^2\, d\pi = \tfrac{1}{2}\int \|\mathbf{x}\|^2\, d\mu + \tfrac{1}{2}\int \|\mathbf{y}\|^2\, d\nu - \int \mathbf{x} \cdot \mathbf{y}\, d\pi$$

$$\overset{a)}{\ge} \tfrac{1}{2}\int \|\mathbf{x}\|^2\, d\mu + \tfrac{1}{2}\int \|\mathbf{y}\|^2\, d\nu - \left(\int \Phi\, d\mu + \int \Phi^*\, d\nu\right), \tag{35}$$

where we use Eq. equation 34 in $a)$.

For $\pi_0 := (\mathrm{Id}, \nabla\Phi)_{\#}\mu$ we have equality in equation 34, hence $\pi_0$ attains the minimum in equation 35. This proves that $\pi_0$ is an optimal Kantorovich coupling. If we restrict to Monge transports $S$ with $S_{\#}\mu = \nu$, then $\pi_S := (\mathrm{Id}, S)_{\#}\mu \in \Pi(\mu, \nu)$ and the same argument shows that

$$\int \tfrac{1}{2}\|\mathbf{x} - S(\mathbf{x})\|^2\, d\mu \ge \int \tfrac{1}{2}\|\mathbf{x} - \nabla\Phi(\mathbf{x})\|^2\, d\mu,$$

i.e. $T = \nabla\Phi$ is optimal among transport maps.

3. If $\mu \ll \mathcal{L}^n$, Brenier's theorem (Villani et al., 2009) guarantees that any optimal plan for the quadratic cost is induced by a unique $\mu$-a.e. map which is the gradient of a convex function. Since $\pi_0$ is optimal and is induced by the map $T$, uniqueness implies that the Brenier (optimal) transport map coincides $\mu$-a.e. with $T = \nabla\Phi$. The $M$-Lipschitz bound follows from equation 29.

$\square$

This theorem shows that, in general, there exist Lipschitz optimal transport maps between multi-modal distributions. However, if the source distribution is supported on a single simply connected domain (for example, a unit disk), while the target distribution is supported on a disjoint union of two or more simply connected domains (for example, two non-intersecting unit disks), then there is no optimal transport map given by the gradient of a strongly convex function.

**Lemma 1.** *Let $\Omega \subset \mathbb{R}^d$ be a nonempty open simply connected set (in particular, connected). Let $\varphi : \Omega \to \mathbb{R}$ be a $C^2$ function which is $m$–strongly convex on $\Omega$, in the sense that*

$$\nabla^2 \varphi(x) \succeq m I_d \quad \text{for all } x \in \Omega$$

*for some constant $m > 0$. Define $T := \nabla\varphi : \Omega \to \mathbb{R}^d$. Then $T(\Omega)$ cannot be written as a disjoint union of $N \geq 2$ nonempty simply connected open sets:*

$$T(\Omega) \neq \bigsqcup_{k=1}^{N} U_k \quad \text{with } N \geq 2, \ U_k \subset \mathbb{R}^d \text{ open, nonempty, simply connected, pairwise disjoint.}$$

*Proof.* Strong convexity implies

$$(\nabla\varphi(x) - \nabla\varphi(y)) \cdot (x - y) \geq m\|x - y\|^2 \quad \text{for all } x, y \in \Omega.$$

If $x \neq y$, the right-hand side is strictly positive, so $\nabla\varphi(x) \neq \nabla\varphi(y)$. Hence $T = \nabla\varphi$ is injective on $\Omega$.

Since $T$ is Lipschitz and therefore continuous, $\Omega$ is connected, and $T$ is injective, the image $T(\Omega)$ is connected (continuous image of a connected set). This contradicts the assumption that $T(\Omega)$ is a union of at least two disjoint nonempty open sets. $\square$

In particular, combining this lemma with Brenier's theorem, we obtain the following implication: if two probability measures have supports given by disjoint unions of open sets with different numbers of connected components (for example, a source supported on a single simply connected domain and a target supported on a disjoint union of several simply connected domains), then the quadratic-cost optimal transport map between them cannot be represented as the gradient of a globally strongly convex potential. Equivalently, the associated Brenier potential cannot be uniformly strongly convex on the whole domain. Strong convexity must necessarily break down near the interfaces separating the target components.

## B  SUPPORTING LEMMAS

**Lemma 2.** *Let $\Phi(\mathbf{x}, t) = \frac{1}{2}\|\mathbf{x}\|^2 + \epsilon\psi_\epsilon(\mathbf{x})$ and $\mathcal{U}(\mathbf{x}, t) = \frac{1}{2}\|\mathbf{x}\|^2 + \epsilon\psi(\mathbf{x})$ with $\psi = -\frac{1}{2}\log\rho$ be the Brenier and heat potentials, respectively, $\rho$ be a distribution, and $\rho_\epsilon$ be its convolution with a Gaussian kernel $\gamma_\epsilon$. For an open ball $B_{2r} \subset K$, there exists a constant $C \equiv C(\rho)$ such that for $\epsilon > 0$, and $|t - \epsilon| < c\epsilon$, the difference $\phi = \psi_\epsilon - \psi$ is bounded by*

$$\|\phi\|_{\mathcal{C}^{2,\alpha}(B_r)} \leq C\epsilon. \tag{36}$$

*Proof.* We consider the lineariztation of the pushforward, giving us

$$\rho(\mathbf{x}) = \rho_\epsilon(T_\epsilon(\mathbf{x}))\det(\nabla T_\epsilon(\mathbf{x}))$$
$$= \rho_\epsilon(\mathbf{x}) + \epsilon\left(\nabla\rho_\epsilon(\mathbf{x}) \cdot \nabla\psi_\epsilon(\mathbf{x}) + \rho_\epsilon(\mathbf{x})\Delta\psi_\epsilon(\mathbf{x})\right) + \mathcal{O}(\epsilon^2), \tag{37}$$

which using the heat expansion

$$\rho_\epsilon = \rho + \frac{\epsilon}{2}\Delta\rho + \mathcal{O}(\epsilon^2)$$

leads us to the approximate FP equation

$$\nabla \cdot (\rho_\epsilon \nabla \psi_\epsilon) = \frac{\rho - \rho_\epsilon}{\epsilon} + \mathcal{O}(\epsilon)$$

$$= -\frac{1}{2}\Delta\rho + \mathcal{O}(\epsilon) \tag{38}$$

on $K$. We now consider the difference in the maps $\phi = \psi_\epsilon - \psi$. Plugging into both sides of 38 gives us

$$-\nabla \cdot (\rho_\epsilon \nabla \phi) = -\nabla \cdot (\rho_\epsilon \nabla (\psi_\epsilon - \psi))$$

$$= \frac{1}{2}(\Delta\rho - \Delta\rho_\epsilon) + \mathcal{O}(\epsilon). \tag{39}$$

Using the linearity of the Laplacian and the first-order expansion from Lemma 3, we obtain

$$\frac{1}{2}\Delta(\rho - \rho_\epsilon) + \mathcal{O}(\epsilon) = -\frac{1}{2}\Delta(\frac{\epsilon}{2}\Delta\rho + \mathcal{O}(\epsilon^2)) + \mathcal{O}(\epsilon). \tag{40}$$

Setting $-\nabla \cdot (\rho_\epsilon \nabla \phi) =: g_\epsilon$, we now have that $\|g_\epsilon\|_{\mathcal{C}^\alpha(B_{2r})} \leq c\epsilon$ for some constant $c > 0$. This defines an elliptic PDE with $\mathcal{C}^\alpha$ coefficients, allowing the use of interior Schauder estimates (Evans, 2022), giving us

$$\|\phi\|_{\mathcal{C}^{2,\alpha}(B_r)} \leq C' \left( \|g_\epsilon\|_{\mathcal{C}^\alpha(B_{2r})} + \|\phi\|_{L^\infty(B_{2r})} \right). \tag{41}$$

The first term is bounded by the discussion above, and we use the Alexandroff-Bakelman-Pucci (ABP) estimate to bound the $L^\infty$-norm over $B_{2r}$ (Lemma 4) to prove the claim. $\square$

**Lemma 3.** *Let $\rho \in \mathcal{C}^4(\mathbb{R}^d; \mathbb{R})$, and $\gamma_t$ the Gaussian density $\mathcal{N}(0, \mathbf{\Sigma}_t)$. Then, we have that*

$$\rho_t(\mathbf{x}) = \rho(\mathbf{x}) + \frac{t}{2}\Delta\rho(\mathbf{x}) + \frac{t^2}{8}\Delta^2\rho(\mathbf{x}) + R_t, \quad \|R_t\|_\infty \leq Ct^3. \tag{42}$$

*Proof.* Using the formulation of $\rho_t(\mathbf{x})$ as an expectation, we apply a fourth-order Taylor expansion of $\rho$ around $\mathbf{x}$:

$$\rho(\mathbf{x} + \sqrt{t}\mathbf{z}) = \rho(\mathbf{x}) + \sum_{|\alpha|=1}^{4} \frac{(\sqrt{t}\mathbf{z})^\alpha}{\alpha!} \partial^\alpha \rho(\mathbf{x}) + R_5(\mathbf{x}, \mathbf{z}, t), \tag{43}$$

where we sum over a multi-index $\alpha$. Then we will average this equation with respect to the Gaussian $\mathbf{z} \sim \mathcal{N}(0, \mathbf{I}_d)$. Due to the Wick's theorem for gaussian random variables, the odd moments will vanish and the even moments will lead to

$$\mathbb{E}\left[ \frac{t}{2} \sum_{i=1}^{d} \mathbf{z}_i^2 \, \partial_{ii}\rho(\mathbf{x}) \right] = \frac{t}{2}\Delta\rho(\mathbf{x}). \tag{44}$$

for the second-order term. The forth-order term will become

$$\mathbb{E}\left[ \frac{t^2}{24} \sum_{i,j,k,\ell} \mathbf{z}_i\mathbf{z}_j\mathbf{z}_k\mathbf{z}_\ell \, \partial_{ijkl}\rho(\mathbf{x}) \right] = \frac{t^2}{8}\Delta^2\rho(\mathbf{x}). \tag{45}$$

So for $\rho_t(\mathbf{x}) = \mathbb{E}_{\mathbf{z} \sim \mathcal{N}(0, \mathbf{I}_d)}[\rho(\mathbf{x} + \sqrt{t}\mathbf{z})]$ we have obtained

$$\rho_t(\mathbf{x}) = \rho(\mathbf{x}) + \frac{t}{2}\Delta\rho(\mathbf{x}) + \frac{t^2}{8}\Delta^2\rho(\mathbf{x}) + R_t(\mathbf{x}), \tag{46}$$

with the remainder term $\|R_t\|_\infty \leq Ct^3$. $\square$

**Lemma 4.** *Let $\phi$ be a solution to $-\nabla \cdot (\rho_\epsilon \nabla \phi) = g_\epsilon$ in $B_{2r}$, where $0 < m < \rho_\epsilon \leq M$ and $\rho_\epsilon \in C^\infty(B_{2r})$. Then there exists a positive constant $C \equiv C(n, \lambda, \Lambda, \|\nabla\rho_\varepsilon\|_{L^\infty(B_{2r})})$ such that*

$$\|\phi\|_{L^\infty(B_{2r})} \leq C \, r^2 \, \|g_\epsilon\|_{L^\infty(B_{2r})}. \tag{47}$$

*Proof.* Let $R := 3r$ and choose $\eta \in C_c^\infty(B_R)$ such that $\eta \equiv 1$ on $B_{2r}$, $|\nabla \eta| \lesssim R^{-1}$ and $|D^2\eta| \lesssim R^{-2}$. Define $u := \eta\phi \in W_0^{1,2}(B_R)$. Then $u$ satisfies

$$-\nabla \cdot (\rho_\epsilon \nabla u) = F := \underbrace{\eta g_\varepsilon}_{F_1} + \underbrace{\rho_\epsilon \nabla\phi \cdot \nabla\eta}_{F_2} + \underbrace{\phi \, \nabla \cdot (\rho_\epsilon \nabla \eta)}_{F_3} \quad \text{in } B_R.$$

By the Alexandroff-Bakelman-Pucci (ABP) maximum principle (Gilbarg & Trudinger, 2001, Theorem 9.1), we have

$$\|u\|_{L^\infty(B_R)} \leq CR\|F\|_{L^d(B_R)}. \tag{48}$$

It remains to estimate $\|F\|_{L^d(B_R)}$. The first term is straightforward:

$$\|F_1\|_{L^n(B_R)} \leq \|\eta\|_\infty \|g_\epsilon\|_{L^d(B_R)} \lesssim R\|g_\epsilon\|_\infty.$$

For $F_2$ and $F_3$ we work on the annulus $A := B_R \setminus B_{2r}$, where $\nabla\eta \neq 0$. A standard (Caccioppoli) energy estimate (Evans, 2022, Chapter 6) yields $\|\nabla\phi\|_{L^2(A)} \lesssim R\|g_\epsilon\|_{L^2(B_R)}$. Applying Hölder scaling[2] yields $\|\nabla\phi\|_{L^d(A)} \lesssim R\|g_\epsilon\|_\infty$. Hence

$$\|F_2\|_{L^d(A)} \leq \|\rho_\epsilon\|_\infty \|\nabla\eta\|_\infty \|\nabla\phi\|_{L^d(A)} \lesssim R^{-1} \cdot (R\|g_\epsilon\|_\infty) \lesssim \|g_\epsilon\|_\infty.$$

For $F_3$, write $\nabla \cdot (\rho_\epsilon \nabla\eta) = \nabla\rho_\epsilon \cdot \nabla\eta + \rho_\epsilon \Delta\eta$. Using a Poincaré inequality on $A$, $\|\phi\|_{L^d(A)} \lesssim R^2\|g_\epsilon\|_\infty$, so

$$\|F_3\|_{L^d(A)} \leq \|\phi\|_{L^d(A)}\Big( \|\nabla\rho_\epsilon\|_\infty \|\nabla\eta\|_\infty + \|\rho_\epsilon\|_\infty \|D^2\eta\|_\infty \Big) \lesssim (R^2\|g_\epsilon\|_\infty)(R^{-1}+R^{-2}) \lesssim R\|g_\epsilon\|_\infty.$$

Collecting, $\|F\|_{L^d(B_R)} \lesssim R\|g_\epsilon\|_\infty$, and hence

$$\|u\|_{L^\infty(B_R)} \lesssim R(R\|g_\epsilon\|_\infty) = R^2\|g_\epsilon\|_\infty.$$

Since $u \equiv \phi$ on $B_{2r}$ and $R = 3r$, the desired bound follows. $\qquad\square$

**Lemma 5** (Wasserstein closeness of cut and uncut). *Let $\rho$ have finite second moment, and $\widehat{\rho}^\eta = Cut_\eta(\rho)$. Then*

$$W_2\big(\rho, \widehat{\rho}^\eta\big) \leq C(\rho)\sqrt{m^\eta}, \qquad m^\eta := 1 - \int_{\{\rho > \eta\}} \rho\, dx,$$

*where $C(\rho)$ depends only on $\int |x|^2 \rho(dx)$.*

*Proof.* Construct a coupling that leaves the mass on $\{\rho > \eta\}$ fixed and transports the tail mass $m^\eta$ into the kept part independently. Using $|x - y|^2 \leq 2|x|^2 + 2|y|^2$ and the second moment bound yields $W_2^2 \leq C m^\eta$. $\qquad\square$

**Lemma 6** (Stability of Brenier potentials). *Let $\mu_k \to \mu$ and $\nu_k \to \nu$ in $W_2$, with $\mu_k, \mu$ absolutely continuous and $\{\mu_k\}, \{\nu_k\}$ tight and with uniformly bounded second moments. Let $\Phi_l, \Phi$ be (normalized) convex Brenier potentials so that $\nabla\Phi_{k\#}\mu_k = \nu_k$ and $\nabla\Phi_\#\mu = \nu$. Then, along a subsequence, $\Phi_k \to \Phi$ locally uniformly (up to additive constants), and $\nabla\Phi_k \to \nabla\Phi$ in measure w.r.t. $\mu$.*

*Proof.* This is standard: $\Gamma$-convergence of the Kantorovich functionals plus uniqueness of the optimal map for absolutely continuous sources; see, e.g. Villani et al. (2009, Chapter 28). $\qquad\square$

**Lemma 7** (Absolute continuity and nontriviality of the cut measure). *Let $\mathcal{L}^d$ denote Lebesgue measure on $\mathbb{R}^d$ and let $\rho \in L^1(\mathbb{R}^d)$ be a probability density, $\rho \geq 0$, $\int_{\mathbb{R}^d} \rho\, dx = 1$. For $\eta > 0$ set*

$$A_\eta := \{x \in \mathbb{R}^d : \rho(x) > \eta\}, \qquad \alpha_\eta := \int_{A_\eta} \rho(x)\, dx.$$

*Assume $\eta < \mathrm{ess}\sup \rho$. Define the* cut *probability measure*

$$\widehat{\mu}_\eta(E) := \frac{1}{\alpha_\eta} \int_E \rho(x)\, \mathbf{1}_{A_\eta}(x)\, dx \qquad \text{for Borel } E \subset \mathbb{R}^d.$$

*Then:*

---

[2] We compute the $L^d$-norm over a set of diameter $r^{-1}R^{d-1}$ which preserves the dominance of $R$.

(i) $0 < \alpha_\eta \leq 1$ and $\widehat{\mu}_\eta$ is a probability measure.

(ii) $\widehat{\mu}_\eta \ll \mathcal{L}^d$, with density

$$\widehat{\rho}_\eta(x) := \frac{\rho(x)\,\mathbf{1}_{A_\eta}(x)}{\alpha_\eta} \in L^1(\mathbb{R}^d).$$

In particular, for every $E$ with $\mathcal{L}^d(E) = 0$ one has $\widehat{\mu}_\eta(E) = 0$.

*Proof.* *(i) Nontriviality.* Since $\eta < \mathrm{ess\,sup}\,\rho$, the set $\{x : \rho(x) > \eta\}$ has positive Lebesgue measure. Hence

$$\alpha_\eta = \int_{A_\eta} \rho\,dx \geq \int_{\{\rho > \eta\}} \rho\,dx \geq \eta\,\mathcal{L}^d(\{\rho > \eta\}) > 0.$$

Trivially $\alpha_\eta \leq \int \rho\,dx = 1$. Therefore $\widehat{\mu}_\eta$ is well-defined and

$$\widehat{\mu}_\eta(\mathbb{R}^d) = \frac{1}{\alpha_\eta} \int_{A_\eta} \rho\,dx = 1.$$

*(ii) Absolute continuity.* For any Borel $E$ with $\mathcal{L}^d(E) = 0$,

$$\widehat{\mu}_\eta(E) = \frac{1}{\alpha_\eta} \int_E \rho\,\mathbf{1}_{A_\eta}\,dx \leq \frac{1}{\alpha_\eta} \int_E \rho\,dx = 0,$$

so $\widehat{\mu}_\eta \ll \mathcal{L}^d$, with density $\widehat{\rho}_\eta = \rho\,\mathbf{1}_{A_\eta}/\alpha_\eta \in L^1$. □

**Corollary 1** (Finite second moment and two–sided bounds on compacts). *Assume in addition that $\int_{\mathbb{R}^d} |x|^2 \rho(x)\,dx < \infty$. Then:*

(a) $\int |x|^2\,d\widehat{\mu}_\eta(x) = \dfrac{1}{\alpha_\eta} \int_{A_\eta} |x|^2 \rho(x)\,dx < \infty.$

(b) *For any bounded Borel set $D \subset \mathbb{R}^d$,*

$$\widehat{\rho}_\eta(x) \geq \frac{\eta}{\alpha_\eta}\ \text{ on } D \cap A_\eta, \qquad \widehat{\rho}_\eta(x) \leq \frac{\|\rho\|_{L^\infty(D)}}{\alpha_\eta}\ \text{ on } D,$$

*whenever $\rho \in L^\infty(D)$. In particular, on any fixed compact $K$ for which $\rho$ is bounded, $\widehat{\rho}_\eta$ enjoys two–sided bounds on $K \cap A_\eta$.*

*Proof.* (a) Immediate from $\alpha_\eta > 0$ and $\int_{A_\eta} |x|^2 \rho \leq \int |x|^2 \rho < \infty$.

(b) The lower bound follows from Lemma 7 (iii). The upper bound is

$$\widehat{\rho}_\eta(x) = \frac{\rho(x)\mathbf{1}_{A_\eta}(x)}{\alpha_\eta} \leq \frac{\|\rho\|_{L^\infty(D)}}{\alpha_\eta} \quad \text{for } x \in D.$$

□

**Corollary 2** (Brenier map exists for the cut source). *Let $\nu$ be any probability measure on $\mathbb{R}^d$ with finite second moment. Under the assumptions of Lemma 7 and Corollary 1 (a), the quadratic-cost Monge problem from $\widehat{\mu}_\eta$ to $\nu$ admits a unique solution given by a* Brenier map*: there exists a convex function $\widehat{\Phi}^\eta$ such that*

$$(\nabla\widehat{\Phi}^\eta)_\#\widehat{\mu}_\eta = \nu,$$

*and $\nabla\widehat{\Phi}^\eta$ is unique $\widehat{\mu}_\eta$-a.e.*

*Proof.* By Lemma 7, $\widehat{\mu}_\eta \ll \mathcal{L}^d$. By Corollary 1 (a), both $\widehat{\mu}_\eta$ and $\nu$ have finite second moments. Brenier's theorem (for the quadratic cost) then guarantees existence and uniqueness of the optimal transport map as the gradient of a convex potential. □

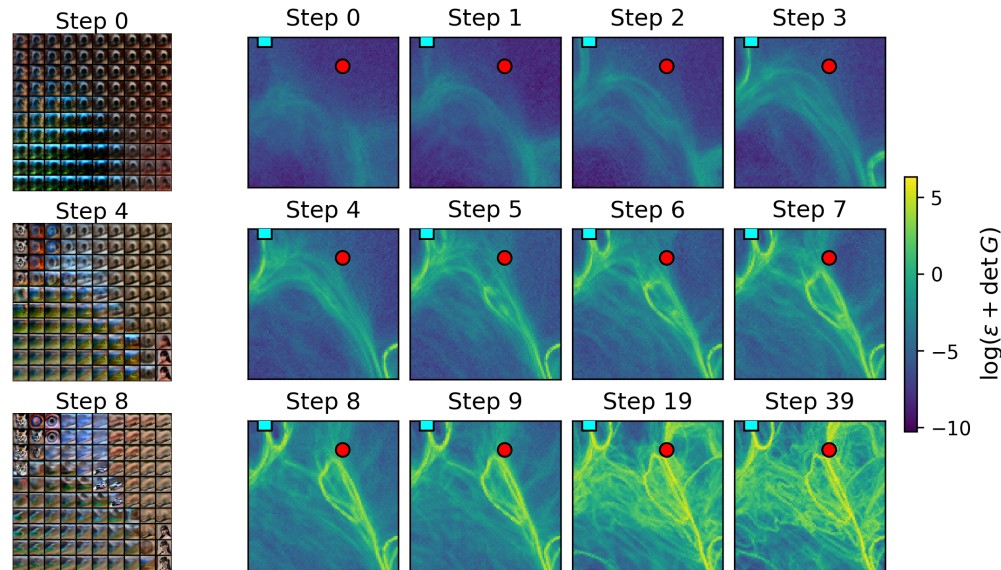

Figure 10: Left: a grid of predicted noiseless images, $\hat{x}_0(\alpha, \beta)$, over a 2D latent slice. Right: evolution of the determinant of the CLIP pullback metric, $G$, shown on a log scale as $\log(\varepsilon + \det G)$ (with $\varepsilon = 10^{-8}$) across denoising timesteps for the same slice. The determinant is proportional to the local volume element of the latent space; high values indicate regions where the features of the generated images change abruptly. The red circle and cyan square mark the two latent points used in Figure 11.

**Lemma 8** (1D approximation forces large slope). *Let $g$ be convex on $(-r, r)$ with $g'(0^+) - g'(0^-) = \Delta > 0$. If $h$ is $\mathcal{C}^1$ on $(-r, r)$ and $\|h - g\|_{L^\infty(-r,r)} \leq \epsilon r \Delta / 8$, then*

$$\sup_{|s| \leq r/2} |h'(s)| \geq \Delta/4.$$

*Proof.* If $\sup |h'| < \Delta/4$ on $[-r/2, r/2]$, then by the mean value theorem the variation of $h$ across $[-r/2, 0]$ and $[0, r/2]$ is too small to follow the two different affine pieces of $g$, contradicting the uniform proximity. A direct integration furnishes the stated constants. $\square$

## C  IMAGE EDITING

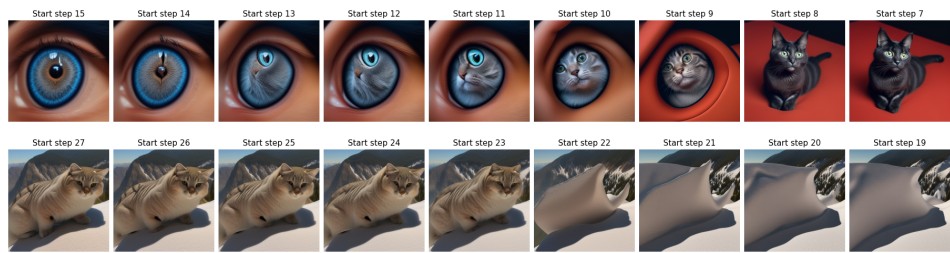

Figure 11: Prompt-guided edits: (top) an eye edited with 'a green eyed cat'; cyan square latent at Figure 1; (bottom) cat in mountains edited with 'snow mountains'; red circle latent at Figure 1

We support our findings also with the following examples. Two images being in one phase, means that their latents are close to each other and predictions of noiseless images are similar. We evaluate prompt-guided editing via partial DDIM inversion (Mokady et al., 2023). Given a source image $x_{\text{src}}$ and source generic prompt, we invert to an intermediate step $t_k$ to obtain a latent state $x_{t_k}$. We

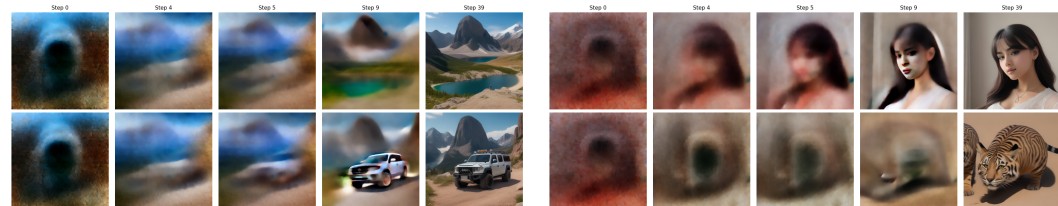

Figure 12: Nearby-grid latent pairs: left $(\alpha, \beta) = (0.51, 0.50)$ vs $(0.51, 0.51)$; right $(\alpha, \beta) = (0.22, 1.00)$ vs $(0.23, 1.00)$. Similarity holds before a boundary; it breaks after.

then resume reverse diffusion conditioned on a target prompt $p_{\text{tgt}}$ to produce an edited image $\hat{x}_0$, holding sampler hyperparameters (guidance scale, steps, $\eta$) fixed across $k$. We observe that editing success rates correlate with mode-splitting times. Qualitative results are shown in Figure 11, where different edits require different numbers of steps: (top) for the cyan point $(\alpha, \beta) = (0.99, 0.11)$ we need to invert back to step 11, while for the red point $(\alpha, \beta) = (0.80, 0.62)$ (bottom) we need only step 23. Notably, this editing also exhibits an abrupt transition. We show in more detail the effects of emerging phase boundaries on geodesic length in latent space in the supplementary materials (Appendix C).

We visualize pairs of nearby points $(\alpha, \beta)$ on the latent grid and their predicted noiseless images. Before two points are separated by a phase boundary (high-$S$ ridge), their latents yield similar noiseless predictions; once a boundary intervenes, the predictions diverge.

# D   CONNECTION BETWEEN THE DETERMINANT OF THE PULLBACK METRIC AND THE LIPSCHITZ CONSTANT OF THE GENERATIVE MAP

In the experimental section we measured the determinant of the pullback metric. Here we establish its relationship with the local Lipschitz constant of the generative map $F$ in order to tie our experimental results with the theoretical analysis. We present this connection with the following lemma.

**Lemma 9.** *High values of square root of the determinant of the pullback metric $\sqrt{\det G(z)}$ imply a large local Lipschitz constant, via the bound:*

$$\mathrm{Lip}(F; z) \geq \left(\sqrt{\det G(z)}\right)^{1/2}. \tag{49}$$

*Proof.* Let $F : \mathbb{R}^2 \to \mathbb{R}^m$ be the generative map, restricted to a latent space slice, which in our case is a map that takes two coordinates on a slice and returns a predicted noiseless image.

Assume that $F$ has a well-defined Jacobian. Fix $z \in \mathbb{R}^2$. Denote the Jacobian of $F$ at $z$ by $J := J_F(z) \in \mathbb{R}^{m \times 2}$ and the induced pullback metric by

$$G(z) = J^\top J \in \mathbb{R}^{2 \times 2}. \tag{50}$$

Let $\sigma_1 \geq \sigma_2 \geq 0$ be the singular values of $J$. By definition, the local Lipschitz constant of $F$ at $z$ is

$$\mathrm{Lip}(F; z) = \sup_{\|v\|=1} \|Jv\|. \tag{51}$$

The right-hand side is exactly the operator norm of $J$,

$$\|J\|_{\mathrm{op}} := \sup_{\|v\|=1} \|Jv\|, \tag{52}$$

which is equal to the largest singular value of the Jacobian. If $\sigma_1 \geq \sigma_2 \geq 0$ are the singular values of $J$, we have

$$\mathrm{Lip}(F; z) = \|J\|_{\mathrm{op}} = \sigma_1. \tag{53}$$

Since $G = J^\top J$, the eigenvalues of $G$ are $\lambda_1 = \sigma_1^2$ and $\lambda_2 = \sigma_2^2$. Therefore,

$$\det G(z) = \lambda_1 \lambda_2 = \sigma_1^2 \sigma_2^2, \tag{54}$$

and hence

$$\sqrt{\det G(z)} = \sigma_1 \sigma_2. \tag{55}$$

Thus

$$\left(\sqrt{\det G(z)}\right)^{1/2} = (\sigma_1 \sigma_2)^{1/2}. \tag{56}$$

Since $\sigma_2 \leq \sigma_1$, we have

$$(\sigma_1 \sigma_2)^{1/2} \leq (\sigma_1 \sigma_1)^{1/2} = \sigma_1 = \mathrm{Lip}(F; z). \tag{57}$$

Therefore

$$\mathrm{Lip}(F; z) \geq \left(\sqrt{\det G(z)}\right)^{1/2}, \tag{58}$$

and we arrive at the statement of this lemma. $\qquad \square$

In other words, this lemma shows that regions where $\sqrt{\det G(z)}$ is large necessarily correspond to regions with a large local Lipschitz constant of the generative map. Since in our experiments we visualize $\sqrt{\det G(z)}$ over two–dimensional latent slices, we are in fact probing the spatial variation of the local Lipschitz constant of the diffusion generative map along these slices. Thus, the ridges and high–value regions observed in our determinant heatmaps can be directly interpreted as high–Lipschitz barriers in the generative map, making our empirical observations quantitatively aligned with the theoretical analysis of Lipschitz growth and phase boundaries.

## D.1 IMAGE RECONSTRUCTION

To test the connection between high Lipschitz constants and DDIM reconstruction, we select 50 points with high metric value (edges; large $\det(G)$ at step 39) and 50 points from basins (low metric value). For each point we perform DDIM inversion (50 steps) and regenerate with identical parameters and prompts. We evaluate reconstruction quality by comparing the regenerated image with the original using four metrics: MSE, SSIM on Canny edges (scikit-image `feature.canny`, $\sigma = 1.5$; classical Canny), CLIP image–image cosine similarity, and DINOv2 image–image cosine similarity. Unless otherwise noted, DDIM inversion uses 50 steps. The sampling locations are shown in Fig. 5, where red points lie near phase boundaries and blue points lie deep inside basins. By Lemma 9, the local Lipschitz constant of the generative map is bounded from below by the pullback metric determinant,

$$\text{Lip}(F; z) \;\geq\; \left( \sqrt{\det G(z)} \right)^{1/2}, \tag{59}$$

so regions with larger $\sqrt{\det G(z)}$ necessarily exhibit larger local Lipschitz constants. The results in Table 2 show that these high-Lipschitz regions (near boundaries) are associated with degraded DDIM inversion reconstruction: MSE increases, while SSIM, CLIP similarity, and DINOv2 similarity all decrease compared to points inside basins.

## E ULTRAMETRIC STRUCTURE OF THE LATENT SPACE

In this section we give a formal statement of how ultrametric property emerges for geodesic distances between points in different basins. Informally, the "basin tree" starts as a single root at early (high-noise) times, when the distribution is effectively unimodal. Each time the generative dynamics undergo a mode-splitting event, a new branch (leaf) is added to this tree, and the resulting inter-basin distances satisfy an ultrametric property in the form of the strong triangle inequality.

**Proposition 3.** *Fix a time $t$ and let $\gamma_t$ be the geodesic distance induced by the pullback metric $G_t$ on the latent space $\mathcal{Z}$. Assume we are in the setting of Theorem 2 and 3: the support decomposes into finitely many Riemannian basins $(\mathcal{B}_{t,1}, \ldots, \mathcal{B}_{t,m})$ separated by singular interfaces $\Sigma_t$, where the JKO transport maps develop Lipschitz blow-up along $\Sigma_t$.*

*Define a distance on basins by*

$$D_t(i,j) \; := \; \min_{\gamma_t : z_1 \to z_2} \; \max_{z \in \gamma} \det G_t(z), \tag{60}$$

*the highest barrier that $\gamma_t$ will have to cross to go from $z_1$ to $z_2$. Then $D_t$ is an ultrametric on the finite set of basins: i.e. for all $i, j, k$, we have the strong triangle inequality*

$$D_t(i,k) \; \leq \; \max\{D_t(i,j), D_t(j,k)\}. \tag{61}$$

*Proof.* Non-negativity and symmetry are always ensured by definition of $D_t(\cdot, \cdot)$. What remains to be proven is that $D_t(i,i) = 0$, $D_t(i,j) = 0$ implies $i = j$, and the strong triangle inequality.

For $i = j$, it is simple to see that $D_t(i,i) = 0$ by choosing $z = z' \in \mathcal{B}_{t,i}$.

Conversely, if $D_t(i,j) = 0$, then it follows from $D_t$'s definition that $z = z'$, implying that $i = j$.[3]

Finally, to show that equation 61 holds. We notice that the hierarchical nesting of basins allows us to state that if $i \neq j \neq k$, then clearly the distance $D_t(i,k)$ is the height of the geodesic $\gamma_t : z_1 \to z_2$, with $z_1 \in \mathcal{B}_{t,i}$ and $z_2 \in \mathcal{B}_{t,k}$. Whether $\gamma_t$ crosses $\mathcal{B}_{t,j}$ or not, $D_t(i,k)$ is dominated by the highest barrier, directly implying equation 61.

$\square$

**Corollary 3.** *We also notice that the ultrametric tree corresponding to $D_t(\cdot, \cdot)$ gains another leaf when the generative dynamics undergo mode-splitting (Theorem 1), as this simply creates a new basin $\mathcal{B}'_t$.*

---

[3]Note that we eliminate the edge-case of $z = z'$ begin on the phase boundary by the measure-zero assumption provided by Theorem 2.

# F ADDITIONAL EXPERIMENTAL RESULTS

We report more experimental results and details in Figures 13, 14, 16, 17.

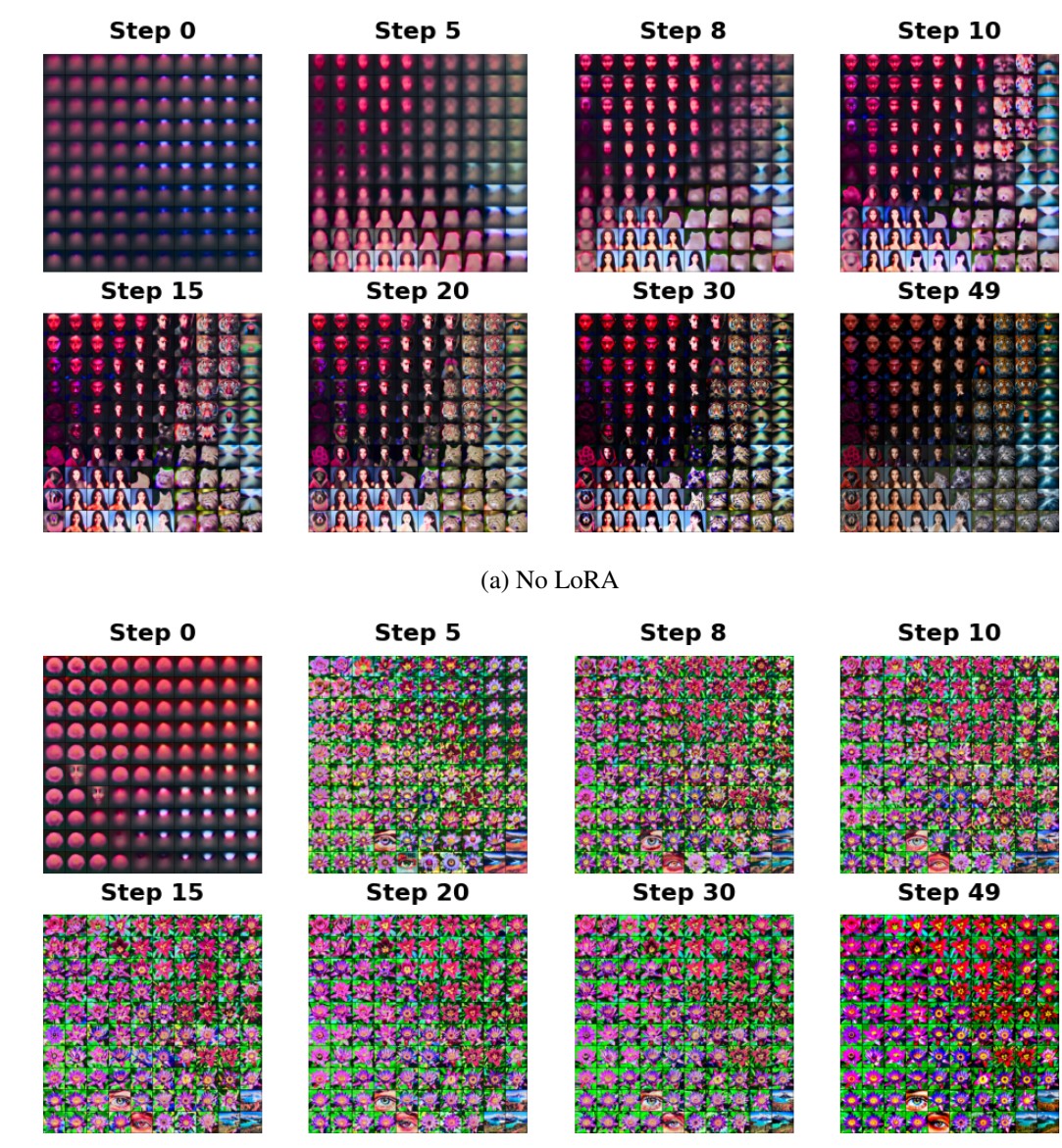

(a) No LoRA

(b) LoRA weight 1.5

Figure 13: Sample grids of generated images (1-frame videos) for the 50-step diffusion process without LoRA (a) and with a LoRA weight of 1.5 (b).

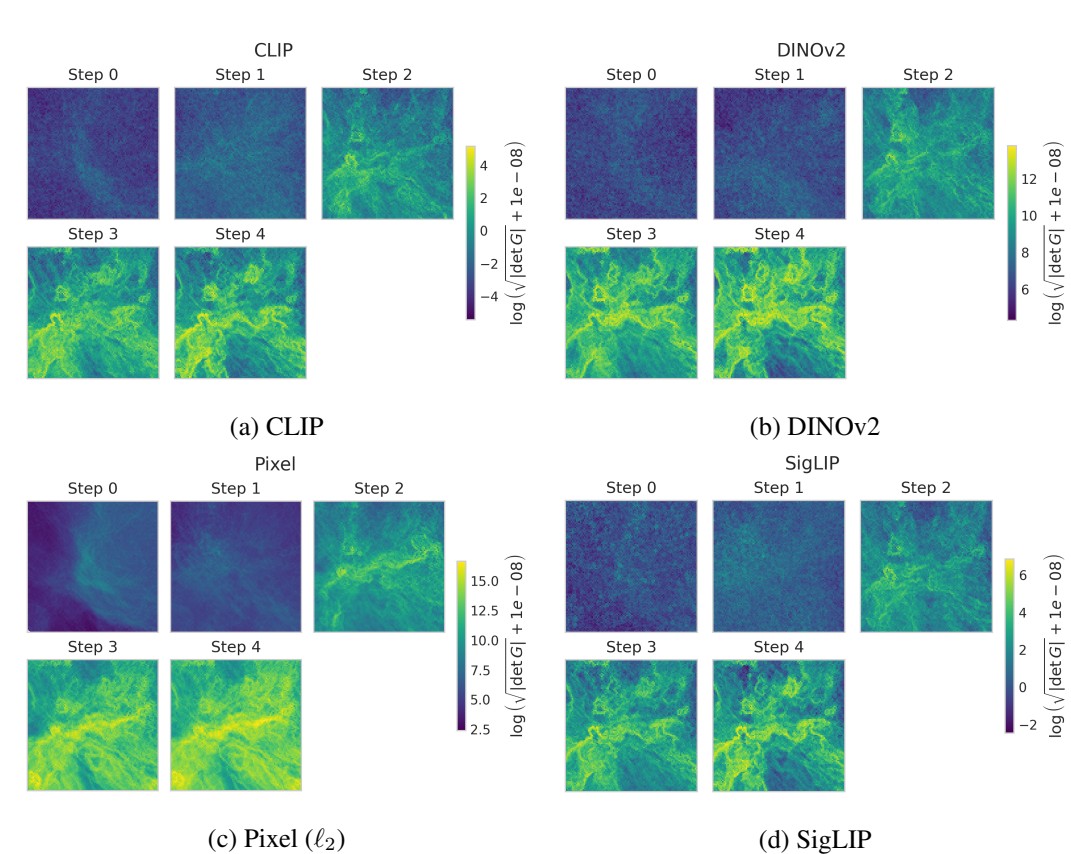

(a) CLIP  (b) DINOv2

(c) Pixel ($\ell_2$)  (d) SigLIP

Figure 14: Final-step $\sqrt{\det G}$ maps across feature extractors (shared color scale). Phase-boundary ridges are consistent across extractors, demonstrating robustness.

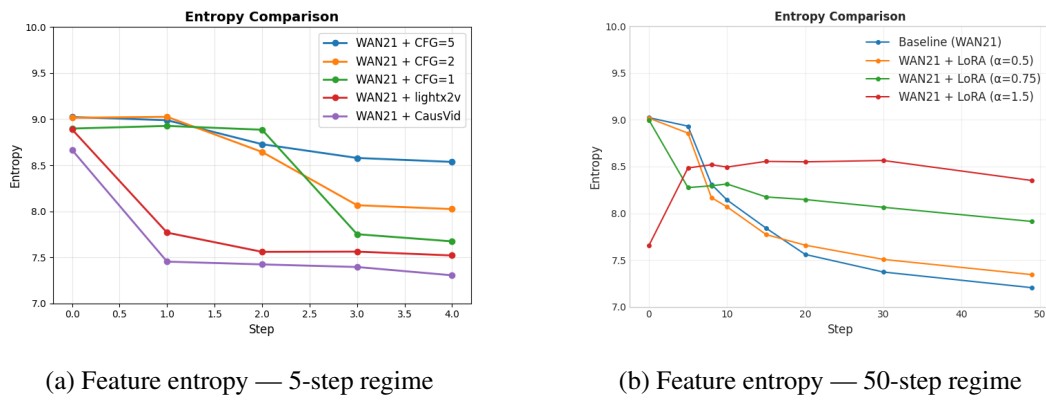

(a) Feature entropy — 5-step regime  (b) Feature entropy — 50-step regime

Figure 15: Feature-space entropy (averaged over the grid) across diffusion steps for two step budgets. Higher entropy correlates with larger average $\sqrt{\det G}$, indicating increased representational dispersion.

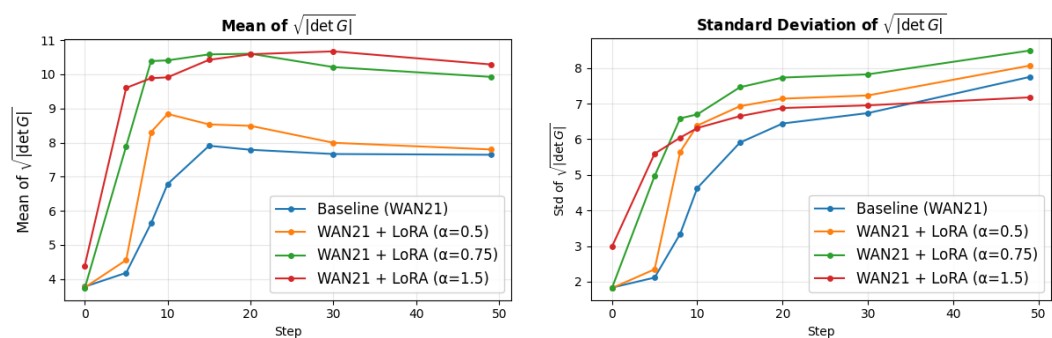

Figure 16: Mean and variance of the metric determinant $\sqrt{\det G}$ computed over the entire latent grid, plotted as a function of the diffusion timestep.

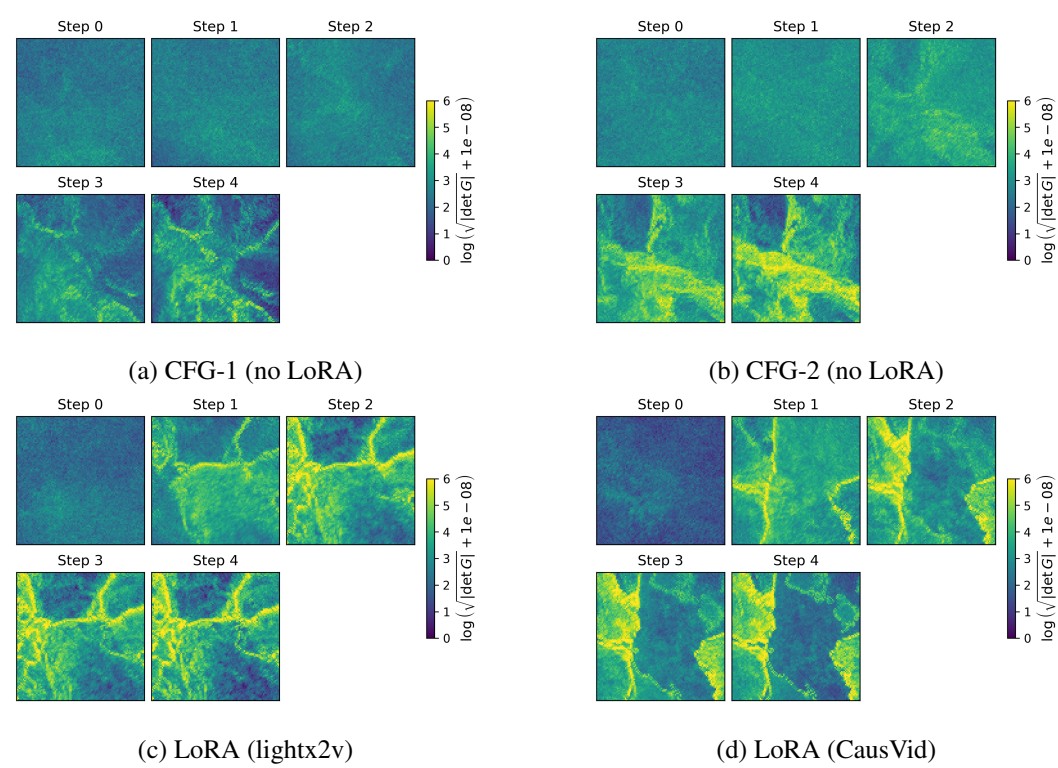

(a) CFG-1 (no LoRA)

(b) CFG-2 (no LoRA)

(c) LoRA (lightx2v)

(d) LoRA (CausVid)

Figure 17: Heatmaps of $\sqrt{\det G}$ computed on DINOv2 for a 5-step regime (K=5) under different classifier-free guidance (CFG) scales and LoRA variants. Panels: (a) CFG-1 (no LoRA); (b) CFG-2 (no LoRA); (c) LoRA (lightx2v); (d) LoRA (CausVid). All panels share the same color scale; bright ridges indicate phase boundaries.

