# OpenReview forum: "Caffarelli Regularity and Hierarchical Phase Boundaries in Diffusion Models"
_ICLR.cc/2026/Conference — Submitted to ICLR 2026_

### Official Review · Reviewer_EKYr · 2025-10-31

**Soundness:** 2
**Presentation:** 2
**Contribution:** 2
**Rating:** 4
**Confidence:** 2

**Summary:**

This paper attempts to formalize the phase transition affect in diffusion model, where at certain reverse diffusion timestep (noisy level), and certain noisy images, feature abruptly appear. The author borrow some interesting theory from physics. And use these idea to create an visualization tool to visualize where these phase transition occur during the diffusion process.

**Strengths:**

I think it's amazing the author is able to track down and visualize phase transition. I'm not really familiar with this literature, but I think this idea of phase transition during reverse diffusion process is only vaguely defined (suddenly an object appear in the reverse diffusion process). The slice latent space + pull back metric and jacobian visualization seems novel.
The author seems to find a deep connection between this ML phenomenon to some theory in physics.
Unfortunately I'm not similar with the theory that they author uses. So I cannot judge how relevant they are.

**Weaknesses:**

From a practitioner point of view, it seems less clear to me how the theory connects to practice.
The author did not write the practical application part clear. What is ($\alpha$, $\beta$), or $\log(\det G)$ has to do with mode-splitting times, and how does that predict the steps for editing images, and how does that depends on the images and edit prompt.

**Questions:**

At high level, how is the theory section related to your experiment, specifically, parametrize slice of latent space, clip pull back metric and determinant of Jacobian measure? How are these related to JKO, Caffarelli’s regularity, etc...

Could you explain the image editing section a little bit more? Staring with a clean image, to edit, you said go back to a specific intermediate time step $x_{t_k}$, meaning you add specific amount of noise. At this point, the noisy image $x_{t_k}$ has different ($\alpha$, $\beta$) and $\log(\det G)$. Could you explain the intuition on how does each parameter, time step $t_k$, ($\alpha$, $\beta$) and $log(\det G)$ affect image editing?

I'm not confident to review this paper but I'm happy to change my score if I have a better understanding of the paper.

---

> ### Author Response · Authors · 2025-11-26
> **Response to Reviewer EKYr**
>
> We thank the reviewer for pointing out where the paper could be clearer. In the revision we tried to address these concerns.
>
>
> **How the theory connects to experiment** The theory predicts that irregularities of diffusion mapping are related to the mode coverage in the diffusion models.
> These irregularities could be formally described in several ways:
> using Lipschitz constant of a diffusion map
> via a pullback metric on a latent space, induced by similarity of generated images
> We establish the relation between them: If the pullback metric determinant is high, then the Lipschitz constant is also high, see Lemma 9. In experiments, we directly compute the pullback metric on a 2D slice (it can be done in higher dimensions, but 2D is convenient for visualization and boundary analysis). We then numerically show that image diversity (mode coverage) indeed correlates with presence of boundaries (irregularities of diffusion map).
>
>
> **Relation to image editing** The irregularities also mean an instability of numerical probability flow ODE solution. We experimentally check this instability using DDIM inversion, which is a base for techniques for image editing. The main goal of inversion experiments is to ensure that phase boundaries found by pullback metric are less numerically stable. To make the presentation more clear, we add the DDIM reconstruction errors in the main text (Fig.5 and Table 2), and image editing is deferred to the appendix.
>
>
> **$(\alpha, \beta)$ parameters** We must clarify that $\alpha$ and $\beta$ are fixed coordinates that parametrize 2D plane inside high-dimensional latent space. Concretely, we pick 3 latent anchors $z_0$, $z_1$, $z_2$ at timestep $t=0$ corresponding to noise and define slice by interpolation $z_0 + \alpha(z_1-z_0) + \beta(z_2-z_0)$ with $\alpha, \beta \in [0,1]$. These coordinates do not change with time.
>
>
> **How the theory connects to practice**  Please refer to the General Response for extended discussion. The Discussion section in the paper was updated in a similar way.

---

### Official Review · Reviewer_w5Pr · 2025-11-01

**Soundness:** 3
**Presentation:** 2
**Contribution:** 3
**Rating:** 6
**Confidence:** 2

**Summary:**

The authors study the emergence of phase-transition-like behavior in diffusion models: small perturbations in latent noise can cause abrupt changes in generated outputs. They propose a theoretical framework based on optimal transport theory (JKO scheme) and Caffarelli regularity, by linking high-Lipschitz regions of the generative map to “phase boundaries” in latent space. The authors support the theory with empirical analyses using pullback metrics derived from perceptual similarity (e.g., CLIP embeddings), showing hierarchical phase boundary formation and its modification under LoRA, diffusion distillation, and classifier-free guidance. Their study suggests that such singularities structure the latent space in a hierarchical, ultrametric fashion, impacting image editing and generative smoothness.

**Strengths:**

1. The authors integrate Caffarelli regularity with the JKO approximation, providing a rigorous mathematical framework for understanding discontinuities in diffusion mappings
2. The authors provide empirical validations via visualizations on Wan 2.1 and Stable Diffusion 1.5
3. The theory has practical intuition for diffusion inversion, editing difficulty, and LoRA effects

**Weaknesses:**

1. Although the paper visualizes phase boundaries via heatmaps of $\sqrt{det(G)}$ and feature entropy, it does not provide quantitative metrics that relate the sharpness or density of these boundaries to concrete model behaviors such as FID, reconstruction error, or editing success rate. This limits the paper's practical insights.
2. It is unclear whether assumption 1 can fully correspond to empirical diffusion behavior, and explanations or supports of the assumption will make the paper sound.

**Questions:**

See weakness.

---

> ### Author Response · Authors · 2025-11-26
> **Response to Reviewer w5Pr**
>
> We thank the reviewer and agree that proposed changes will improve the paper quality. In the revision we implemented most of the recommendations.
>
> **Phase boundaries and Reconstruction error**
> To assess how the local Lipschitz value results in reconstruction error, we sample points from the phase boundaries and from basins (see Fig.5). For each point we perform DDIM inversion procedure and regenerate images, comparing final result with the initial sample via MSE, SSIM on canny image, and cosine similarity of CLIP and DINOv2 embeddings. Boundary points show higher reconstruction errors than basin points across all metrics, Table 2.
>
> **Table 2**: *Quantitative metrics for DDIM inversion (mean ± std).*
>
> | Metric (mean±std) | Boundary | Basin |
> |-------------------|----------|-------|
> | MSE ↓             | 0.035 ± 0.001 | 0.020 ± 0.001 |
> | SSIM ↑            | 0.618 ± 0.004 | 0.740 ± 0.003 |
> | CLIP ↑            | 0.711 ± 0.008 | 0.839 ± 0.007 |
> | DINOv2 ↑          | 0.397 ± 0.015 | 0.683 ± 0.014 |
>
> We also investigate how the metric grids behave combined with LoRAs in various settings.
>
> **Sharpness and density of phase boundaries vs. Diversity** We estimate sharpness and density of boundaries and relate these metrics to diversity of generated samples for Wan2.1, Table 1. Table 1 shows quantitatively that both the sharpness and the number of boundaries decrease when the acceleration LoRAs are applied. At the same time, the diversity of generated images is reduced, indicating worse mode coverage. This statement is valid for 50 steps and for a 5-step accelerated schedule.
>
>  **Table 1**: *Wan 2.1. Numerical evaluation of Fig.4 from the paper. Dataset statistics for different acceleration LoRA weights.
> Edge count: number of pixels detected as edges by Canny on a 150×150 grayscale image.
> Avg. sharpness: mean gradient magnitude using 5×5 Sobel filters.
> Image diversity: average cosine distance between CLIP or SigLIP embeddings over 5×10⁴ random image pairs.*
> | Model                | Edge count | Avg. sharpness | CLIP Diversity ↑        | SigLIP Diversity ↑      |
> |----------------------|------------|----------------|--------------------------|---------------------------|
> | *50 steps*           |            |                |                          |                           |
> | base model           | **5502**   | **1462.3**     | **0.4151 ± 0.0007**      | **0.3411 ± 0.0005**       |
> | lightx2v, w=0.50     | _5451_     | _1388.5_       | _0.3729 ± 0.0007_        | _0.3144 ± 0.0006_         |
> | lightx2v, w=0.75     | 3897       | 1038.0         | 0.2607 ± 0.0006          | 0.2469 ± 0.0005           |
> | lightx2v, w=1.50     | 1045       | 562.6          | 0.0989 ± 0.0004          | 0.1103 ± 0.0004           |
> | *5 steps*            |            |                |                          |                           |
> | lightx2v, w=1.0      | 3851       | 1025.7         | 0.3323 ± 0.0007          | 0.3114 ± 0.0006           |
> | CausVid, w=1.0       | 2317       | 794.2          | 0.1970 ± 0.0005          | 0.2064 ± 0.0006           |
>
> **Correspondence of Assumption 1 to empirical diffusion behavior** Assumption 1 follows the 'Union of Manifolds Hypothesis' [1] and suggests that the real data distribution is supported on disjoint sets. The paper by Brown et al. empirically verifies this hypothesis on common image datasets, and in our experiments, we do not observe contradicting facts. The equivalence of Assumption 1 and ‘Union of Manifolds Hypothesis’ was added in the revised version.
>
> [1] Brown, Bradley CA, et al. "Verifying the Union of Manifolds Hypothesis for Image Data." The Eleventh International Conference on Learning Representations. 2022.

---

### Official Review · Reviewer_18L9 · 2025-11-01

**Soundness:** 3
**Presentation:** 3
**Contribution:** 2
**Rating:** 6
**Confidence:** 1

**Summary:**

This work proposes a theoretically grounded way to understand where and how the phase boundaries arise in the latent space of diffusion models. The experiments are performed to understand how phase boundaries evolve. The impact of CFG and LoRA on phase boundaries is studied. Further visual experiments are validatied on 2D latent slices of images.

**Strengths:**

* Interesting findings that both classifier-free guidance (CFG) and LoRA increase the average geometric sensitivity of the slice
* Highlights how distilled models, CGF, LoRA degrade mode coverage
* The study is based on Caffarelli’s theory and combine it with the Jordan Kinderlehrer-Otto (JKO) scheme for the Fokker-Planck equation.
* Figure 2 nicely illustrates the idea of cut-trick and the problems with existing models.

**Weaknesses:**

I had struggle understanding how this analysis helps develop better generative models or how it can improve existing diffusion models.

**Questions:**

The work is mainly theoretical and is beyond my mathematical skills. I see the problems raised by the paper and how they propose a nice theoretically sound way to address these issues. I am somehow left wondering what is this study be useful for? How can we design a model that can build on top of the theoretical guarantees promised in this work?
I believe it is very important to answer these questions, at least, somewhat partially. This would greatly help adoption of the ideas in practice.

---

> ### Author Response · Authors · 2025-11-26
> **Response to Reviewer 18L9**
>
> We are grateful to the Reviewer for acknowledging the theoretical soundness. We agree that the paper will benefit from extended discussion of practical implications. Therefore, we revised the abstract and Discussion to highlight the outcomes of our theoretical and experimental analysis.
>
> **How to design better models**. From our analysis, it follows that singularities are inherent to diffusion mappings when real image data is generated from unimodal Gaussian noise. We observe such behavior in SD1.5 and Wan2.1 during 50-step generation, which supports the theory. However, non-Lipschitzness is often seen as not a desirable property: for instance, earlier works [1][2][3] smooth the latent space, effectively reducing the spatial Lipschitz constant. Our findings suggest that these constraints will suppress model expressive power and should be avoided.
>
> **How to design better distillation**. The second practical observation is related to DMD distillation and acceleration LoRAs. Intuitively, a low-rank addition to model weights should not change the expressive power of a base model. Despite that, we numerically show that LoRAs reduce mode coverage. From our analysis, a possible explanation is that few-step generation does not accumulate singularities and is not able to reach a high Lipschitz constant. Better distillation is, therefore, may be connected with increasing the non-Lipschitzness of a diffusion mapping for few-step generation.
>
> [1] Guo, Jiayi, et al. "Smooth diffusion: Crafting smooth latent spaces in diffusion models." Proceedings of the IEEE/CVF Conference on Computer Vision and Pattern Recognition. 2024.
>
> [2] Hahm, Jaehoon, et al. "Isometric representation learning for disentangled latent space of diffusion models." Proceedings of the 41st International Conference on Machine Learning. 2024.
>
> [3] Zhou, Yifan, et al. "Alias-free latent diffusion models: Improving fractional shift equivariance of diffusion latent space." Proceedings of the Computer Vision and Pattern Recognition Conference. 2025.

---

### Official Review · Reviewer_Rrfo · 2025-11-04

**Soundness:** 3
**Presentation:** 3
**Contribution:** 3
**Rating:** 6
**Confidence:** 2

**Summary:**

This paper explains why diffusion models exhibit phase boundaries in their latent spaces, where nearby latent points can produce drastically different outputs. The authors analyze the reverse probability flow of diffusion models through the lens of JKO (Jordan-Kinderlehrer-Otto) gradient flow and show that the generative process can be understood as a sequence of optimal transport maps. Using Caffarelli regularity theory, they prove that when the data distribution is multimodal, the generative map must develop non-smooth regions (singular sets) separating different modes. These regions manifest as sharp ridges in the pullback metric computed from perceptual feature embeddings such as CLIP. Experiments on Stable Diffusion and Wan 2.1 confirm that these boundaries emerge hierarchically along the reverse diffusion trajectory. The paper further shows that LoRA fine-tuning and distillation shift and simplify these boundaries, explaining the observed fidelity&diversity trade off in practical text2image generation.

**Strengths:**

1. Applying the JKO scheme and Caffarelli regularity theory to the analysis of phase boundaries in diffusion models is a new and theoretically sound contribution.
2. The paper connects the theoretical analysis and the experimental results in a clear way, providing a unified explanation of how phase boundaries form.
3. The robustness analysis using different feature extractors supports that the observed phase boundaries are not dependent on a particular embedding choice.

**Weaknesses:**

1. The cut trick is central to the paper, but its methodological justification is not sufficiently explained. It is unclear whether this operation preserves the essential structure of the phase boundaries or introduces artificial artifacts. A sensitivity analysis of the cut threshold and a comparison with alternative approaches would be necessary.
2. The claim regarding an ultrametric structure is supported primarily by 1D experiments, and there is no clear evidence that the same structure holds in high-dimensional image spaces. It is likely that the behavior may differ in higher dimensions.
3. The paper does not provide quantitative metrics for the image editing experiments. Relying solely on qualitative figures makes the argument less convincing.

**Questions:**

How is OT implemented in practice?
How was the cut-trick parameter \eta selected?
What is the finite difference step size used in the Jacobian estimation?
What are the LoRA rank and α values?
What sampler and hyperparameters were used?

---

> ### Author Response · Authors · 2025-11-26
> **Response to Reviewer Rrfo**
>
> We thank the reviewer for the feedback and following the suggestion of this review, we added quantitative metrics for the DDIM inversion and other experiments.
>
> **Sensitivity analysis of ‘cut trick’**.  We should clarify that a cut trick is a theoretical tool that allows to apply Cafarelly theory to the reverse diffusion process. To avoid the possible confusion, the “cut trick” was renamed to Cut-off operator, Fig.2. It was not implemented in the sampling process.
>
> **Ultrametric Structure in high dimensions**. We agree that the ultrametric claim was not further depicted beyond the 1D case in Figure 6. We have formalized the implicit statement regarding the ultrametricity in the supplementary material (Proposition 3, Appendix E) and discuss how this claim generalizes to any dimension for multimodal data distributions.
>
> **Hyperparemeters** The model utilizes two LoRA adapters for diffusion acceleration, both of rank 32 [1][2]. All LoRA (\alpha) values are set to (1), except if stated otherwise, like in Figure 4 where (\alpha) values are noted in the captions. We use the standard DDIM sampler for SD1.5 and  UniPCMultistepScheduler for Wan2.1. We use classifier-free guidance with the default CFG scale of (5), unless stated otherwise.
>
> **Table 2**: *Quantitative metrics for DDIM inversion (mean ± std).*
> Boundary points show higher reconstruction errors than basin points across all metrics.
> | Metric (mean±std) | Boundary | Basin |
> |-------------------|----------|-------|
> | MSE ↓             | 0.035 ± 0.001 | 0.020 ± 0.001 |
> | SSIM ↑            | 0.618 ± 0.004 | 0.740 ± 0.003 |
> | CLIP ↑            | 0.711 ± 0.008 | 0.839 ± 0.007 |
> | DINOv2 ↑          | 0.397 ± 0.015 | 0.683 ± 0.014 |
>
>
>
> [1] https://huggingface.co/Kijai/WanVideo_comfy/blob/main/Wan21_T2V_14B_lightx2v_cfg_step_distill_lora_rank32.safetensors
>
> [2] https://huggingface.co/Kijai/WanVideo_comfy/blob/main/Wan21_CausVid_14B_T2V_lora_rank32.safetensors

---

### Author Response · Authors · 2025-11-26
**General Response**

We are grateful to the reviewers for valuable feedback and appreciation of the theoretical novelty of our paper. However, concerns were raised regarding the practical outcomes and the quantitative evaluation and metrics. To address these points, we revised our paper, updating the Method and Discussion sections; the abstract was changed accordingly and now emphasises the practical contribution.

The revision aims to strengthen theoretical claims by providing supporting experiments with numerical metrics:
- quantitative estimation of the sharpness and number of phase boundaries, (Table 1 and Table 2)
- diversity of generated images, calculated with CLIP and SigLIP (Table 1)
- a set of metrics related to DDIM reconstruction error  (Table 2)

Despite the theoretical nature of this paper, the statements regarding the Lipschitz constant have practical significance, now highlighted in the Discussion.

First, non-Lipschitzness of diffusion mapping is often seen as an undesirable property: for instance, earlier works [1][2][3] smooth the latent space, effectively reducing the spatial Lipschitz constant. However, our findings suggest that a high Lipschitz constant is a property of unimodal-to-multimodal diffusion mapping, and phase boundaries (regions with high Lipschitz constant) are associated with mode splitting.

The second practical observation is related to DMD distillation and acceleration LoRAs. Intuitively, a low-rank addition to model weights should not change the expressive power of a base model. Despite that, we numerically show that LoRAs reduce mode coverage. This may be attributed to the DMD loss or the few-step generation itself. We finish the discussion with an open question: could it be that sequential calls of a model allow a greater Lipschitz constant of a mapping, due to its accumulation? This, in turn, implies that for a few-step generation we may need to increase the non-Lipschitzness of a model.

**Table 1**: *Wan 2.1. Numerical evaluation of Fig. 4. Dataset statistics for different acceleration LoRA weights.
Edge count: number of pixels detected as edges by Canny on a 150×150 grayscale image.
Avg. sharpness: mean gradient magnitude using 5×5 Sobel filters.
Image diversity: average cosine distance between CLIP or SigLIP embeddings over 5×10⁴ random image pairs.*
| Model                | Edge count | Avg. sharpness | CLIP Diversity ↑        | SigLIP Diversity ↑      |
|----------------------|------------|----------------|--------------------------|---------------------------|
| *50 steps*           |            |                |                          |                           |
| base model           | **5502**   | **1462.3**     | **0.4151 ± 0.0007**      | **0.3411 ± 0.0005**       |
| lightx2v, w=0.50     | 5451     | 1388.5       | 0.3729 ± 0.0007        | 0.3144 ± 0.0006         |
| lightx2v, w=0.75     | 3897       | 1038.0         | 0.2607 ± 0.0006          | 0.2469 ± 0.0005           |
| lightx2v, w=1.50     | 1045       | 562.6          | 0.0989 ± 0.0004          | 0.1103 ± 0.0004           |
| *5 steps*            |            |                |                          |                           |
| lightx2v, w=1.0      | 3851       | 1025.7         | 0.3323 ± 0.0007          | 0.3114 ± 0.0006           |
| CausVid, w=1.0       | 2317       | 794.2          | 0.1970 ± 0.0005          | 0.2064 ± 0.0006           |

**Table 2**: *Quantitative metrics for DDIM inversion (mean ± std).
Boundary points show higher reconstruction errors than basin points across all metrics.*
| Metric (mean±std) | Boundary | Basin |
|-------------------|----------|-------|
| MSE ↓             | 0.035 ± 0.001 | 0.020 ± 0.001 |
| SSIM ↑            | 0.618 ± 0.004 | 0.740 ± 0.003 |
| CLIP ↑            | 0.711 ± 0.008 | 0.839 ± 0.007 |
| DINOv2 ↑          | 0.397 ± 0.015 | 0.683 ± 0.014 |

[1] Guo, Jiayi, et al. "Smooth diffusion: Crafting smooth latent spaces in diffusion models." Proceedings of the IEEE/CVF Conference on Computer Vision and Pattern Recognition. 2024.

[2] Hahm, Jaehoon, et al. "Isometric representation learning for disentangled latent space of diffusion models." Proceedings of the 41st International Conference on Machine Learning. 2024.

[3] Zhou, Yifan, et al. "Alias-free latent diffusion models: Improving fractional shift equivariance of diffusion latent space." Proceedings of the Computer Vision and Pattern Recognition Conference. 2025.

---

### Meta-Review · Area_Chair_mhqx · 2026-01-01

**Summary:**

This paper proposes a novel perspective on the latent geometry of diffusion models by linking their generative processes to optimal transport (OT) regularity results. The core idea of connecting Caffarelli-type results to phase boundaries in diffusion processes is original and holds significant potential for understanding these complex generative models. However, the current manuscript is not fully ready, due to a lack of clarity and rigorous definitions of key concepts. Key OT concepts like Wasserstein gradient flows, JKO schemes, and Brenier maps are introduced without proper background or intuition, making the central argument difficult to follow even for technically inclined readers. Furthermore, the use of inconsistent notation for critical variables makes the paper difficult to follow.

Beyond the presentational issues, the paper suffers from several mathematical and experimental limitations.
(i) The mathematical analysis relies on several oversimplified assumptions, such as treating the forward marginal as a simple heat kernel convolution in a setting with time-varying drift and diffusion. The justification for approximating the reverse ODE with JKO steps is also flawed, as the premises regarding fixed potential gradient flows do not hold.
(ii) Experimentally, the paper relies almost exclusively on qualitative visual evidence to support claims about "phase boundaries" and "hierarchical geometry." It lacks crucial quantitative metrics, rigorous statistical testing, and essential implementation details, making the results non-reproducible and the conclusions speculative.

Due to these substantial theoretical and empirical shortcomings, the paper is not ready for publication.

**Reviewer Concerns:**

(i) The mathematical analysis relies on several oversimplified assumptions, such as treating the forward marginal as a simple heat kernel convolution in a setting with time-varying drift and diffusion. The justification for approximating the reverse ODE with JKO steps is also flawed, as the premises regarding fixed potential gradient flows do not hold.


(ii) Experimentally, the paper relies almost exclusively on qualitative visual evidence to support claims about "phase boundaries" and "hierarchical geometry." It lacks crucial quantitative metrics, rigorous statistical testing, and essential implementation details, making the results non-reproducible and the conclusions speculative.

**Reviewer Scores:**

none

---

### Decision · Program_Chairs · 2026-01-26

Reject